# Fourier Features in Reinforcement Learning with Neural Networks

**David Brellmann**                                    *david.brellmann@ensta-paris.fr*
*U2IS, ENSTA Paris, Institut Polytechnique de Paris*

**David Filliat**                                        *david.filliat@ensta-paris.fr*
*U2IS, ENSTA Paris, Institut Polytechnique de Paris*

**Goran Frehse**                                        *goran.frehse@ensta-paris.fr*
*U2IS, ENSTA Paris, Institut Polytechnique de Paris*

**Reviewed on OpenReview:** *https://openreview.net/forum?id=LWotmCKC6Y*

## Abstract

In classic Reinforcement Learning (RL), encoding the inputs with a Fourier feature mapping is a standard way to facilitate generalization and add prior domain knowledge. In Deep RL, such input encodings are less common since they could, in principle, be learned by the network and may therefore seem less beneficial. In this paper, we present experiments on Multilayer Perceptrons (MLP) that indicate that even in Deep RL, Fourier features can lead to significant performance gains in both rewards and sample efficiency. Furthermore, we observe that they increase the robustness with respect to hyperparameters, lead to smoother policies, and benefit the training process by reducing learning interference, encouraging sparsity, and increasing the expressiveness of the learned features. However, a major bottleneck with conventional Fourier features is that the number of features increases exponentially with the state dimension. As a remedy, we propose a simple, light version that only has a linear number of features yet empirically provides similar benefits. Our experiments cover both shallow/deep, discrete/continuous, and on/off-policy RL settings.

## 1 Introduction

In classic Reinforcement Learning (RL), the performance of algorithms depends critically on the way the states of the system are represented as features. Choosing appropriate features for a task is an important way of adding prior domain knowledge since cleverly distributing information into states facilitates generalization. For linear function approximations, the representation is usually hand-designed according to the task at hand and projected into a higher-dimensional space to facilitate linear separation (Sutton & Barto, 2018). In RL, such feature encodings for linear function approximation have been proposed in the form of, e.g., Polynomial Features (Lagoudakis & Parr, 2003) or Tile Coding (Albus, 1971). However, the main bottleneck of these feature encodings is that they do not scale to high-dimensional inputs as they grow exponentially in size with the input dimension.

In recent years, interest in Deep RL has grown in part because Neural Networks (NN) are able to learn feature representations. This allows algorithms to learn complex tasks from raw sensory data without prior knowledge (Mnih et al., 2015; Schulman et al., 2017; Lillicrap et al., 2015; Haarnoja et al., 2018). In Deep Learning, it is common to apply min-max normalization (Bishop et al., 1995) or batch normalization (Ioffe & Szegedy, 2015). However, preprocessing inputs with hand-designed features are less common since such features could, in principle, be learned by the network and thus may seem less beneficial. In recent work, it has been shown that preprocessing inputs with Random Fourier Features (Rahimi & Recht, 2007) help Multilayer Perceptrons (MLP) to control the frequencies that the network tends to learn first (Tancik et al.,

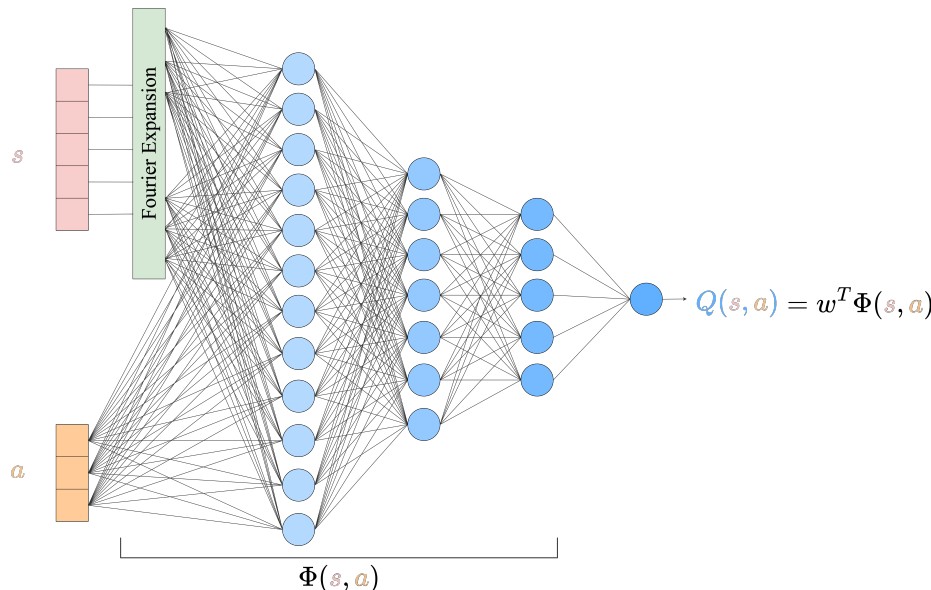

Figure 1: **Example of a 3-layer MLP with preprocessing via Fourier Expansion**. The state $\boldsymbol{s}$ is processed with a Fourier function expansion before being passed into the MLP with the action $\boldsymbol{a}$. Learned features $\phi(\boldsymbol{s}, \boldsymbol{a})$ are used for the prediction of the Q-value $Q(\boldsymbol{s}, \boldsymbol{a})$.

2020; Wang et al., 2021) and improves the training performance for NN (Mehrkanoon & Suykens, 2018; Mitra & Kaddoum, 2021). In Deep RL, it has been observed that Tile Coding can improve performance, sample efficiency, and robustness to hyperparameter variations by mitigating learning interference (Ghiassian & Huizhen Yu, 2018; Ghiassian et al., 2020; Liu et al., 2019). In this paper, we empirically study preprocessing inputs with a function expansion based on the Fourier series in Deep RL, as illustrated in Figure 1. The study is based on kinematic observation-based benchmarks, where observations are expressed as state vectors whose components are the agent's kinematic quantities. Although the advantages of Fourier Features have been investigated in the case of standard RL (Konidaris et al., 2011), to the best of our knowledge, their use has not yet been studied in deep RL. Concurrent work proposes tuning the scale of learnable Fourier features as one means of ensuring that high-frequency components of the value function are captured (Li & Pathak, 2021; Yang et al., 2021). Another recent work studied the use of the Fourier series with MLP in computer vision (Benbarka et al., 2022).

Our main contributions in this paper can be summarized as follows:

- While Fourier Features are standard in classic Reinforcement Learning, we suggest that **Fourier Features are beneficial in kinematic observation-based** RL problems with Neural Networks. We observe significant performance gains in both rewards and sample efficiency and extend the range of usable hyperparameters. In our experiments, Fourier Features outperform other common types of input preprocessing.

- We **empirically investigate the effects of Fourier features on the learning process** and show that Fourier features lead to smoother Neural Networks, mitigate learning interference, promote sparsity, and increase the expressivity of learned features.

- We propose a **light, scalable version of Fourier Features** to avoid the exponential explosion of traditional Fourier Features while maintaining much of their benefits.

The remainder of the paper is structured as follows. We first discuss some of the limitations of Neural Networks in Deep RL. Then, we present our Fourier Light Feature encoding and demonstrate its advantages. Finally, we analyze in more detail the underlying reasons for these gains.

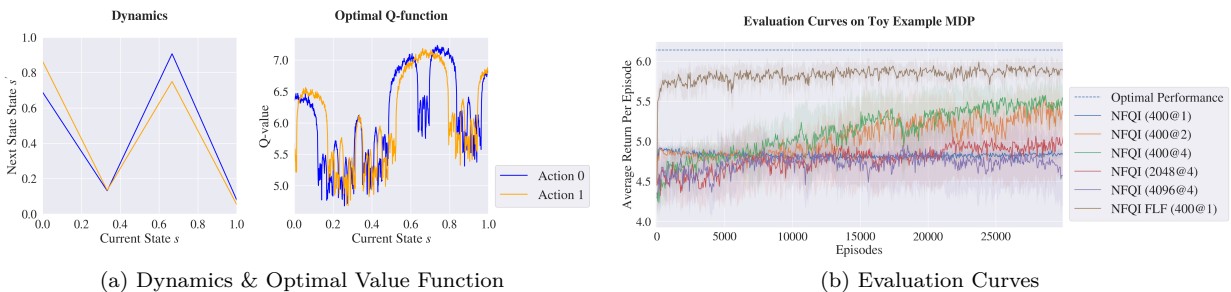

(a) Dynamics & Optimal Value Function
(b) Evaluation Curves

Figure 2: Example of a Toy MDP with simple forward dynamics and complex value function, adapted from Dong et al. (2020). **MLPs without function expansion underperform on the toy MDP.** Evaluation curves are averaged over 10 training runs. Shading indicates the 95% confidence interval (CI). The tested architectures are 1-layer MLP with 400 hidden neurons, 2-layer MLP with 400 hidden neurons, and 4-layer MLP with 400, 2048, and 4096 hidden neurons.

## 2 Limitation of the Expressiveness of Neural Networks in Deep RL

Neural networks have seen widespread success in deep learning, so one might expect that they perform equally well in reinforcement learning. This, however, is not always the case. A study by Dong et al. (2020) suggests that Neural Networks have limitations when predicting Q-functions in RL. The authors show, both theoretically and empirically, that there are many Markov Decision Processes (MDPs) whose optimal Q-functions and policies can be very complex with high-frequency components and variations, even in the case of simple dynamics in a one-dimensional continuous state space. Such complexities arise from the recursive applications of the Bellman optimality operator.

To illustrate the limitations of Neural Networks for approximating value functions in Deep RL, we will compare the evaluation curves and the Q-function of different MLP architectures, with and without Fourier Features, using the following toy MDP distribution from Dong et al. (2020). The state space is $\mathcal{S} = [0, 1)$, the discrete action space is $\{0, 1\}$, and the reward is proportional to the state, i.e., $r(s) = s$. The dynamics are randomly sampled from the space of piece-wise linear functions with a fixed number $k$ of "kinks." For our example, shown in Figure 2a, we use $k = 2$ and uniformly sample the value of each kink in this dynamic function between 0 and 1. Figure 2a also shows the optimal Q-function (computed with dynamic programming), which is complex and has both high- and low-frequency components.

To facilitate the analysis, we use the Neural Fitted Q-Iteration (FQI) algorithm from Riedmiller (2005). The basic idea underlying Neural FQI is the following: instead of updating the neural value function online, the update is performed offline on a set of transition experiences. During the $k$-th fitting iteration, FQI trains the Q-function, $Q_k$, to match the target values, $y_k = R + \gamma P^\pi Q_{k-1}$, which are generated using the previous Q-function, $Q_{k-1}$. Figure 3 shows the value function obtained for different MLP architectures. All of them underfit the optimal value function and fail to learn high-frequency components, even in cases that are over-parameterized. Indeed, the spectral bias of Neural Networks towards low-frequency functions, i.e., functions that vary globally without local fluctuations, has been studied in detail in (Rahaman et al., 2019; Bietti & Mairal, 2019; Cao et al., 2021; Xu et al., 2022). Such a bias may prevent MLPs from correctly learning high-frequency components of Q-functions that are complex and have high-frequency components, like the one in our example. This phenomenon is further exacerbated by the fact that Deep RL algorithms based on TD errors generalize poorly and memorize experiences as they are trained (Lyle et al., 2021; 2022; Nikishin et al., 2022). Since low-frequency components are learned first by MLPs, they struggle to learn higher-frequency components. This problem can be alleviated by adding a functional extension to the MLP, such as Fourier features. Using Fourier features in the above example improves the performance and adds high-frequency components, as shown in Figure 3 and Figure 4. In the next sections, we will focus on a specific functional expansion based on the Fourier Series to add high-frequency components to the predictions of Neural Networks.

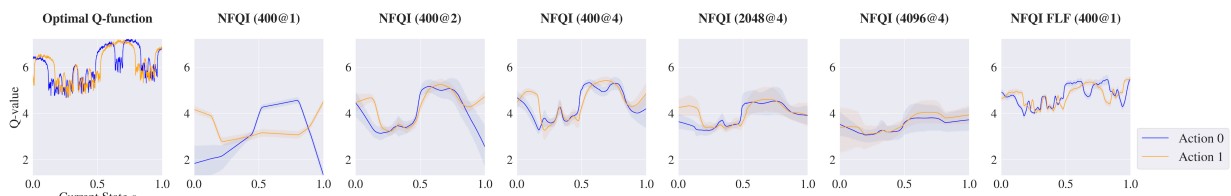

Figure 3: **MLPs without function expansion underfit in comparison to the optimal value function on the toy MDP from Figure 2a**. Results are averaged over 10 training runs. Shading indicates the 95% confidence interval (CI). The tested architectures are 1-layer MLP with 400 hidden neurons, 2-layer MLP with 400 hidden neurons, and 4-layer MLP with 400, 2048, and 4096 hidden neurons.

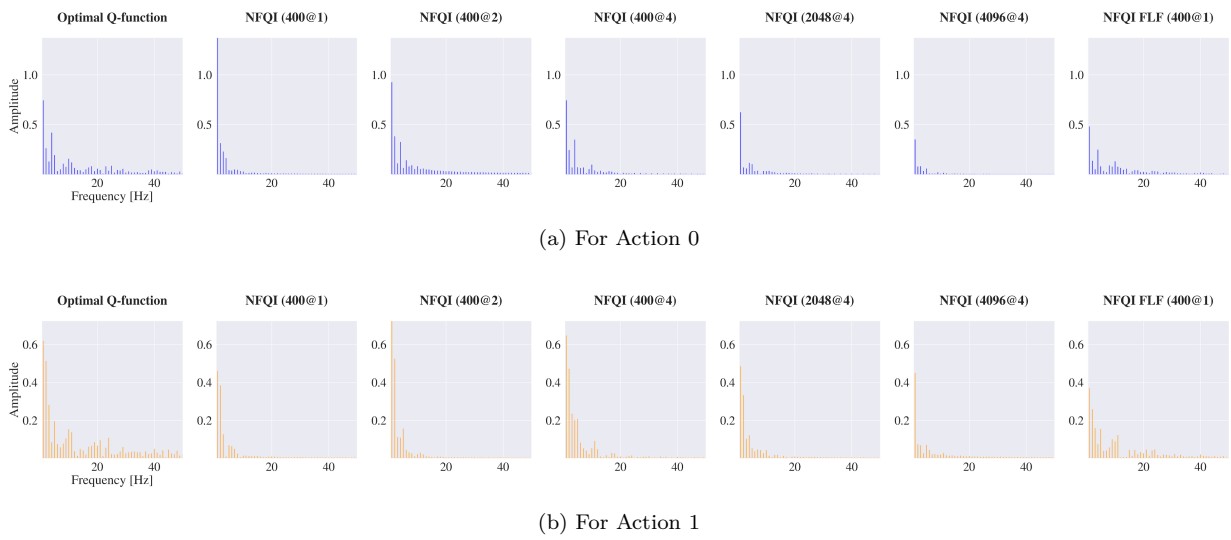

Figure 4: Fourier Spectrum of the averaged predicted Q-function on 10 training runs depicted in Figure 3. The tested architectures are 1-layer MLP with 400 hidden neurons, 2-layer MLP with 400 hidden neurons, and 4-layer MLP with 400, 2048, and 4096 hidden neurons. Those Fourier spectrum do not include the Fundamental frequency since it is much higher than other frequencies.

## 3 Applying Fourier Features to Reinforcement Learning with Neural Networks

**Fourier Features (FF)** were introduced in RL by Konidaris et al. (2011) to approximate linear functions by using the terms of the multivariate Fourier series as features. In practice, Fourier features are easy to use and perform better for linear function approximation than other popular feature encodings such as Tile Coding or Polynomial Basis (Konidaris et al., 2011). Formally, they are generated by the **order-$m$ Fourier Feature** function expansion FF : $[0, 1]^n \rightarrow \mathbb{R}^p$, mapping a normalized state $\boldsymbol{s} \in \mathbb{R}^n$ into a $p$-dimensional feature space (Konidaris et al., 2011), with $p = (m + 1)^n$. For $1 \leq i \leq p$, the feature $i$ is given by

$$\text{FF}_i(\boldsymbol{s}) = \cos(\pi \boldsymbol{s}^\top \boldsymbol{c}^i), \tag{1}$$

where each coefficient vector $\boldsymbol{c}^i$ takes a value in $\{0, \ldots, m\}^n$ (one-to-one). Examples of FF are given in Figure 5.

The inner product $\boldsymbol{s}^\top \boldsymbol{c}^i$ determines the frequency along dimension $i$ and creates interactions between state variables. The major bottleneck of Fourier features is that their dimension grows exponentially with the dimension $n$ of the state space. To remedy this, we propose the following subset of Fourier features that do not join state variables during preprocessing and scale linearly with the state dimension.

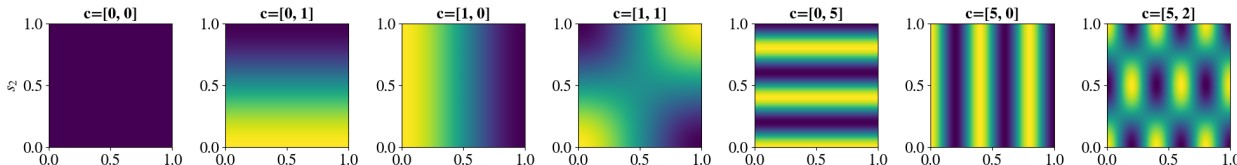

Figure 5: Example of Fourier Features over 2 variables. Darker colors indicate a value closer to 1, and lighter colors indicate a value closer to $-1$. Note that $c = [0, 0]$ results in a constant function. When $c = [0, k_y]$ or $[k_x, 0]$ for positive integers $k_x$ and $k_y$, the basis function depends on only one of the variables, with the value of the non-zero component determining frequency. Only when $c = [k_x, k_y]$ does it depend on both; this basis function represents an interaction between the two state variables. The ratio between $k_x$ and $k_y$ describes the direction of the interaction, while their values determine the basis function's frequency along each dimension.

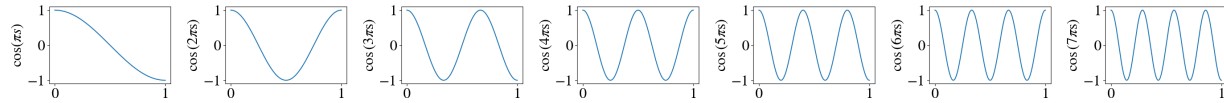

Figure 6: Example of Fourier Light Features.

We call **order-$m$ Fourier Light Features** (FLF) the $n(m+1)$ Fourier Features generated by the **order-$m$ Fourier Light Feature** functional expansion FLF : $[0,1]^n \to \mathbb{R}^{n(m+1)}$ mapping a normalized state $\boldsymbol{s} = [s_1, \ldots, s_n]^T \in [-1, 1]^n$ into a $n(m+1)$-dimensional feature space using a subset of decorrelated Fourier Features as

$$\mathrm{FLF}(\boldsymbol{s}) = [\mathrm{FF}(s_1)| \ldots |\, \mathrm{FF}(s_n)]^T, \tag{2}$$

with $\mathrm{FF}(s_i) = [1, \cos(\pi s_i), \ldots, \cos(m \pi s_i)]\ \ \forall i \in [1, m]$. The choice not to mix state variables in this version of Fourier features is motivated by the fact that Fourier features are not directly used for predictions but rather serve as a preprocessing for the data injected into the NN. We let Neural Networks choose how state variables will be mixed during the learning of the features that will be used for predictions. Examples of order-7 FLF are shown in Figure 6.

**Overall Performance** We apply Fourier Features (FF-NN) and Fourier Light Features (FLF-NN) on the off-policy Deep-Q Network (DQN) algorithm (Mnih et al., 2015) for the discrete action environments and on the on-policy Proximal Policy Optimization (PPO) algorithm (Schulman et al., 2017) for continuous action environments and compare to the algorithms without encoding (NN). Both discrete and continuous environments provide kinematic observations, i.e. observations are expressed as a state vector whose components are the agent's kinematics quantities. For the normalization of the states, we compute the min/max normalization when state variables are bounded and a nonlinear normalization based on arctan otherwise. All algorithms are applied to Neural Network with the ReLu activation function. For computation time reasons and fair comparison, only the learning rate and Fourier order are re-optimized for FF-NN and FLF-NN. Other hyperparameters are taken from Stable Baselines Zoo Raffin (2020). See the appendix for further experimental details. Figure 7 shows the averaged returns per episode for DQN on four discrete-action environments from OpenAI Gym (Brockman et al., 2016). Figure 8 shows the averaged returns per episode of PPO on five continuous-action control tasks from Mujoco (Todorov et al., 2012). For the latter, we only test FLF because the number of standard Fourier Features explodes due to the higher state dimension. In all discrete tasks, except for the LunarLander-v2 task, FLF and FF enhance sample efficiency, i.e. FLF and FF have better performance in terms of cumulative rewards with fewer environment interactions. In the LunarLander-v2 task, their performance does not deteriorate. It is worth noting that in the MountainCar-v0 task, FLF and FF significantly increase the final cumulative reward, as traditional Neural Networks in this experiment are unable to converge to the optimal policy. The increase of final cumulative reward for FLF and FF is not observed in other discrete environments, as they are relatively simple. In all continuous tasks, FLF considerably outperforms the baseline in terms of both cumulative rewards and sample efficiency.

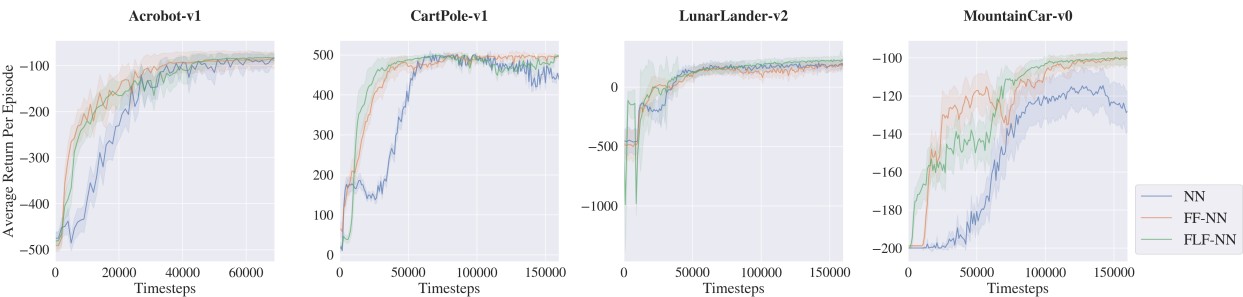

Figure 7: **The use of Fourier features improves performance and sample efficiency of DQN on discrete control tasks.** Evaluation learning curves of NN (blue), FF-NN (orange), and FLF-NN (green), reporting episodic return versus environment timesteps. Results are averaged over 30 training (different seeds), with shading indicating the 95% confidence interval (CI).

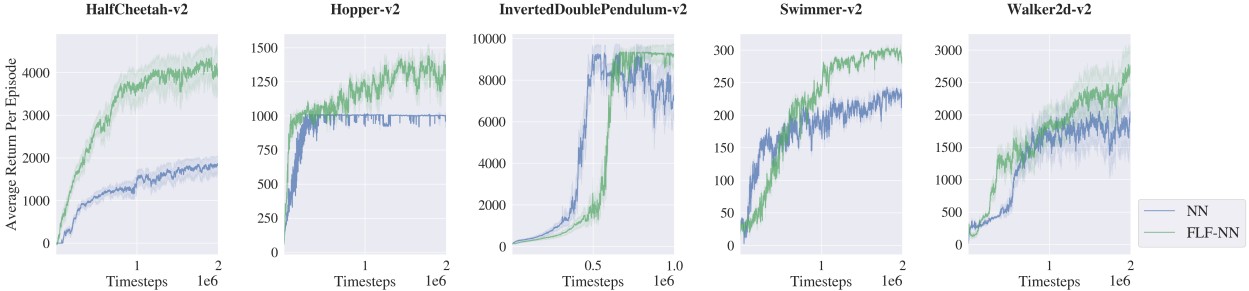

Figure 8: **The use of Fourier Light features improves the performance and sample efficiency of PPO on continuous control tasks.** Evaluation learning curves of NN (blue) and FLF-NN (green), reporting episodic return versus environment timesteps. Results are averaged over 10 training with shading indicating the 95% confidence interval (CI).

Where the dimension is small enough so that we can apply FF, we obtain similar performance by FF and FLF.

**Robustness to Hyperparameter Changes** RL algorithms can be very sensitive to hyperparameter changes (Henderson et al., 2018; Islam et al., 2017). The following experiments indicate that Fourier features reduce the sensitivity to hyperparameters. Figure 9a illustrates how the performance varies with the learning rate, keeping other hyperparameters constant. FF-DQN and FLF-DQN require a smaller learning rate, but perform well over a larger range compared to raw inputs. Experience replay buffers (Lin, 1992; Mnih et al., 2015) and target networks (Mnih et al., 2015) were introduced in RL to mitigate interference problems and have become critical in the training of many deep RL algorithms including DQN. However, it is at the cost of higher computational and memory costs and slower offline learning (Plappert et al., 2018). Zhang & Sutton (2017) highlighted difficulties in properly tuning the buffer size where either too small or too big buffer can have a negative effect on performance. In Figure 9b and 9c we vary only the buffer size or target update frequency while keeping other hyperparameters fixed. In the cases where standard DQN shows large performance variations for different buffer sizes and frequencies, we observe that FF-DQN is both better and less sensitive. This indicates a more stable learning process, with potentially less interference (see also Section 4.3), and makes Fourier Features even more interesting for nonstationary and online problems.

## 4 Observed Effects on Training Neural Networks

In this section, we look at different metrics in order to investigate why Fourier features help the Neural Network to learn better and faster. We study the effects of Fourier features on the sparsity and expressiveness of the NN, which in turn can reduce catastrophic interference. Catastrophic interference occurs

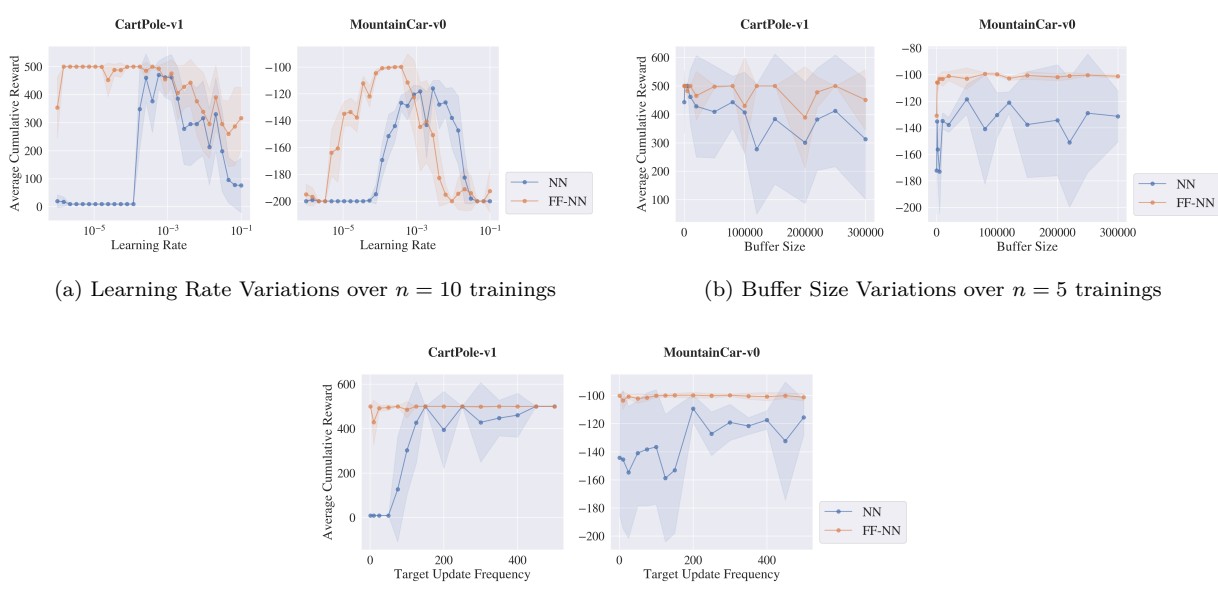

(a) Learning Rate Variations over $n = 10$ trainings

(b) Buffer Size Variations over $n = 5$ trainings

(c) Target Update Variations over $n = 5$ trainings

Figure 9: **Fourier Features are more robust to learning rate, buffer size and target update frequency.** Cumulative reward over different hyperparameter variations, for NN (blue) and FF-NN (orange) on MountainCar-v0 and CartPole-v1. Results are averaged over 10 trainings and shading indicating the 95% confidence interval (CI).

when the learner "forgets" what it has learned in the past by overwriting previous updates to better fit the learned function to recent data (McCloskey & Cohen, 1989; French, 1991). Such interference can significantly slow down learning and even prevent the network from converging to an optimal solution. In sparse representations, only a few features are active (nonzero) for any given input, so each update only impacts a few weights and is less likely to interfere with other updates (Liu et al., 2019; Hernandez-Garcia & Sutton, 2019; Ghiassian et al., 2020; Pan et al., 2020). Another beneficial effect of sparsity is the promotion of locality, where similar inputs should produce similar features. It may be thus easier for the agent to make accurate predictions for an explored local region as the local dynamics are likely to be a simpler function than the global dynamics. The authors showed that learning sparse representations improves performance (Liu et al., 2019; Hernandez-Garcia & Sutton, 2019; Ghiassian et al., 2020; Pan et al., 2020). Functional expansion may help the learning of sparse representations as it was observed by Ghiassian et al. (2020) who empirically shows that applying Tile Coding before passing data into a Neural Network improves the learning of sparse features. Enforcing sparsity can also promote expressiveness through the identification of key attributes by encouraging the input to be well-described by a small subset of attributes. To achieve good performance, a NN needs to extract expressive and fine-grained local features. This is particularly true when consecutive raw inputs are similar and small differences between inputs may lead to different actions. Kumar et al. (2020); Luo et al. (2020); Lyle et al. (2021) identified in RL an implicit under-parameterization of Neural Networks where under-parameterization is defined as an excessive aliasing of learned features, i.e., learned features are mapped into a much smaller subspace than the feature space that could be generated by the NN. Consequently, they observed the neural network behaves as an under-parameterized network, generates less rich features, and leads to poorer performance.

## 4.1 Sparsity of Learned Representations

We measure sparsity with two proxy measures: normalized overlap and instance sparsity (Liu et al., 2019; Hernandez-Garcia & Sutton, 2019; Pan et al., 2020). In the following, we take into account the number of dead neurons, i.e. neurons that have a zero response value for any input, and call the other $m$ neurons

Table 1: **Fourier features and Fourier Light features promote sparsity on discrete control tasks**. Sparsity scores with percentage of dead neurons, normalized activation overlap and instance sparsity obtained for DQN fed with raw inputs (NN), Fourier features (FF-NN) and Fourier Light features (FLF-NN), averaged across environment timesteps. Averages are taken across all timesteps and margins of error of the 95% confidence interval (CI) are computed over 30 trainings. Lower sparsity scores are better and better scores are in **bold**.

| Architecture | | MountainCar-v0 | Acrobot-v1 | CartPole-v1 | LunarLander-v2 |
|---|---|---|---|---|---|
| NN | Dead Neurons | $0.47 \pm 0.09$ | **0.0** | $0.07 \pm 0.02$ | 0.0 |
| | Normalized Overlap | $0.72 \pm 0.08$ | $0.49 \pm 0.04$ | $0.63 \pm 0.04$ | $0.30 \pm 0.01$ |
| | Instance Sparsity | $0.78 \pm 0.07$ | $0.64 \pm 0.02$ | $0.66 \pm 0.03$ | $0.46 \pm 0.02$ |
| FF-NN | Dead Neurons | **0.0** | 0.01 | **0.0** | **0.0** |
| | Normalized Overlap | $\mathbf{0.37 \pm 0.06}$ | $\mathbf{0.05 \pm 0.02}$ | $\mathbf{0.52 \pm 0.02}$ | $\mathbf{0.23 \pm 0.03}$ |
| | Instance Sparsity | $\mathbf{0.57 \pm 0.05}$ | $\mathbf{0.13 \pm 0.04}$ | $\mathbf{0.60 \pm 0.02}$ | $\mathbf{0.40 \pm 0.02}$ |
| FLF-NN | Dead Neurons | **0.0** | **0.0** | **0.0** | **0.0** |
| | Normalized Overlap | $0.43 \pm 0.10$ | $0.16 \pm 0.02$ | $0.79 \pm 0.07$ | $0.39 \pm 0.04$ |
| | Instance Sparsity | $0.62 \pm 0.08$ | $0.30 \pm 0.03$ | $0.85 \pm 0.06$ | $0.55 \pm 0.04$ |

alive. In our experiments, we use NNs with ReLU activation function, wherein dead neurons may occur. Let $d$ be the number of neurons in the penultimate layer. The normalized activation overlap, as proposed by (Hernandez-Garcia & Sutton, 2019), is

$$\text{overlap}(\phi(\boldsymbol{s}_1, a_1), \phi(\boldsymbol{s}_2, a_2)) = \frac{1}{m} \sum_{i=1}^{d} \mathbf{1}_{\phi_i(s_1, a_1) > 0 \wedge \phi_i(s_2, a_2) > 0}. \tag{3}$$

When the normalized overlap between two representations is zero, there is no interference between their corresponding inputs. The normalization with the number of neurons alive avoids misleadingly low scores in cases where only a few neurons are alive. The instance sparsity is the percentage of active units in the feature vector $\phi(\boldsymbol{s}, a)$ for a given input value $(\boldsymbol{s}, a)$ (Liu et al., 2019). We estimate these sparsity measures during the training over the same learned feature matrix $\boldsymbol{\Phi}(\mathcal{D}) = (\phi(\boldsymbol{s}_1, a_1), \dots, \phi(\boldsymbol{s}_b, a_b))^T$ every $1,000$ environment timesteps, where $\mathcal{D} := \{(\boldsymbol{s}_1, a_1), \dots, (\boldsymbol{s}_b, a_b)\}$ is a datatset of $b = 3,000$ state-action pairs. State-action pairs in $\mathcal{D}$ are drawn i.i.d from rollouts obtained with sub-optimal pre-trained policies and random policies. This construction of $\mathcal{D}$ aims to cover state-action pairs likely to be used during the DQN learning. Consequently, our estimations of the percentage of dead neurons are less restrictive than the true percentage of dead neurons. Nevertheless, we believe that measuring sparsity scores over $\mathcal{D}$ makes more sense since it removes neurons only active in parts of the state space that are less likely to be visited by the agent. Our results are summarized in Table 1; the corresponding curves as a function of environment steps can be found in Appendix C. In all tasks, FF results in lower (and thus better) normalized overlap and instance sparsity. There are no dead neurons in FF/FLF, suggesting a better use of the Neural Network capacity. However, FLF increases the sparsity in one instance (CartPole-v1), even though the learning performance of FLF-NN is better than NN in all instances. Hence, sparsity does not seem to be the only beneficial effect of FF/FLF.

### 4.2 Expressiveness of Learned Representations

To measure expressiveness, we compute the effective rank $\text{srank}_\delta$ of the learned feature matrix $\boldsymbol{\Phi}$ built on the set $\mathcal{D}$, normalized by the number of neurons in the penultimate layer like (Kumar et al., 2020). This estimates the proportion of the sum of the $k$ highest singular values $\sigma_1(\boldsymbol{\Phi}) \geq \dots \geq \sigma_{\min(b,d)}(\boldsymbol{\Phi}) \geq 0$ of $\boldsymbol{\Phi}$ that capture $1 - \delta$ (usually $\delta = 0.01$) of the sum of all singular values:

$$\text{srank}_\delta(\boldsymbol{\Phi}) = \frac{1}{d} \min \left\{ k : \frac{\sum_{i=1}^{k} \sigma_i(\boldsymbol{\Phi})}{\sum_{i=1}^{\min(b,d)} \sigma_i(\boldsymbol{\Phi})} \geq 1 - \delta \right\}, \tag{4}$$

where $b$ is the number of state-action pairs used to build $\boldsymbol{\Phi}$ and $d$ is the number of neurons in the penultimate layer of the NN. Intuitively, this quantity represents the number of "effective" unique components of the

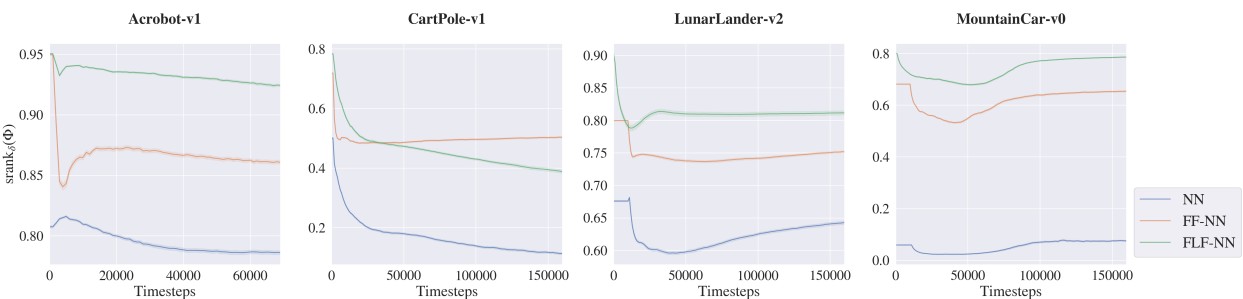

Figure 10: **The use of Fourier features and Fourier Light features enhances the expressiveness of the learned features on discrete control tasks.** Learning curves of the normalized effective rank $\text{srank}_\delta(\mathbf{\Phi})$ for NN fed with raw inputs (blue), Fourier features (orange), and Fourier Light features (green), averaged over 30 trainings with shading indicating the 95% CI.

feature matrix $\mathbf{\Phi}$ that form the basis for linearly approximating the targets. When the network aliases inputs by mapping them to a smaller subspace, $\mathbf{\Phi}$ has only a few active singular directions and $\text{srank}_\delta(\mathbf{\Phi})$ takes thus a small value. Several works pointed to an implicit under-parametrization with the measure of low effective rank during the training. They explained this phenomenon in RL because direct and accurate targets are not available Kumar et al. (2020); Luo et al. (2020); Lyle et al. (2021). We must approximate them with bootstrapping, i.e. by sequentially fitting outputs to target value estimates generated from the function learned in previous iterations. Such targets can not serve as efficient guidance for NNs to extract expressive representations. Figure 10 shows the normalized effective rank over the environment timesteps of training. The learned features are more expressive for FF/FLF in all instances, which may induce a better use of the network capacity and explain better performance. These results are consistent with the absence of dead neurons for FF/FLF reported in Table 1. Features learned with FLF-NN are more expressive than those with FF-NN in most instances. In Deep RL, because targets are estimated with boostrapping, a decrease in the effective rank of learned features can lead to a sequence of NNs with potentially decreasing expressivity and results in degenerate behaviors, generalization problems and drops in performance (Kumar et al., 2020; Luo et al., 2020; Lyle et al., 2021). In Figure 10, the curves with FF/FLF and raw data have a similar general trend in time, but in most instances, the decrease in the effective rank is less pronounced with FF/FLF. This suggests a more stable learning process with less catastrophic interference for FF-NN and FLF-NN.

### 4.3 Learning Interference

The learning interference for a state-action pair $(s, a)$ at time $t$ with parameters $\mathbf{W}$ and loss $L$ is defined as (Lopez-Paz & Ranzato, 2017; Riemer et al., 2018):

$$LI_t = L(\mathbf{W}_{t+1}; \boldsymbol{s}, a) - L(\mathbf{W}_t; \boldsymbol{s}, a).$$

Assuming a small learning rate $\eta$ and using the Taylor series expansion on the loss $L$, we have for an update made to $\mathbf{W}$ with $(\boldsymbol{s}_t, a_t)$:

$$
\begin{aligned}
LI_t &\approx \nabla_{\mathbf{W}} L(\mathbf{W}_t; \boldsymbol{s}_t, a_t)^T (\mathbf{W}_{t+1} - \mathbf{W}_t) \\
&= -\eta \nabla_{\mathbf{W}} L(\mathbf{W}_t; \boldsymbol{s}_t, a_t)^T \nabla_{\mathbf{W}} L(\mathbf{W}_t; s, a)
\end{aligned}
\tag{5}
$$

where the quantity $\nabla_{\mathbf{W}} L(\mathbf{W}_t; \boldsymbol{s}_t, a_t)^T \nabla_{\mathbf{W}} L(\mathbf{W}_t; \boldsymbol{s}, a)$ is the gradient alignment (Bengio et al., 2020; Lopez-Paz & Ranzato, 2017; Riemer et al., 2018; Schaul et al., 2019). The positiveness or negativeness of this quantity determines whether the update is constructive (i.e. positive generalization) or destructive (i.e. interference) on $(\boldsymbol{s}, a)$. To investigate interference, we estimate the stiffness of the gradient alignment (Fort et al., 2020):

$$\rho(\boldsymbol{s}, a, \boldsymbol{s}', a') = \cos(\nabla_{\mathbf{W}} L(\mathbf{W}; \boldsymbol{s}, a), \nabla_{\mathbf{W}} L(\mathbf{W}; \boldsymbol{s}', a')), \tag{6}$$

with parameters $\mathbf{W}$, loss $L$, and cosine similarity $\cos(\boldsymbol{u}, \boldsymbol{v}) = \boldsymbol{u}^T \boldsymbol{v} / ||\boldsymbol{u}|| ||\boldsymbol{v}||$. Based on stiffness, we consider three proxy measures for gradient interference evaluated on the DQN experience replay buffer $\mathcal{B}$:

Table 2: **Fourier features and Fourier Light features mitigate learning interference on discrete control tasks**. Interference measures with Average of Stiffness (AS), Average of Interference (AI), and Interference Risk (IR) averaged across all timesteps for DQN fed with raw inputs (NN), Fourier features (FF-NN), and Fourier Light features (FLF-NN) on discrete control tasks. The symbol ↓ (↑) indicates that a lower (higher) score is better. Best interference measures are in **bold.**

| Architecture | | | MountainCar-v0 | Acrobot-v1 | CartPole-v1 | LunarLander-v2 |
|---|---|---|---|---|---|---|
| NN | AS | ↓ | 0.24 | 0.09 | 0.22 | 0.06 |
| | AI | ↑ | $-0.83$ | $-0.60$ | $-0.92$ | $-0.56$ |
| | IR | ↑ | $-0.91$ | $-0.92$ | $-0.99$ | $-0.94$ |
| FF-NN | AS | ↓ | 0.10 | **0.03** | **0.05** | 0.06 |
| | AI | ↑ | $\mathbf{-0.47}$ | $\mathbf{-0.37}$ | $\mathbf{-0.73}$ | $\mathbf{-0.49}$ |
| | IR | ↑ | $\mathbf{-0.87}$ | $\mathbf{-0.80}$ | $-0.98$ | $\mathbf{-0.92}$ |
| FLF-NN | AS | ↓ | **0.05** | 0.04 | **0.05** | **0.04** |
| | AI | ↑ | $-0.54$ | $\mathbf{-0.38}$ | $-0.86$ | $-0.67$ |
| | IR | ↑ | $\mathbf{-0.87}$ | $\mathbf{-0.79}$ | $-0.98$ | $-0.94$ |

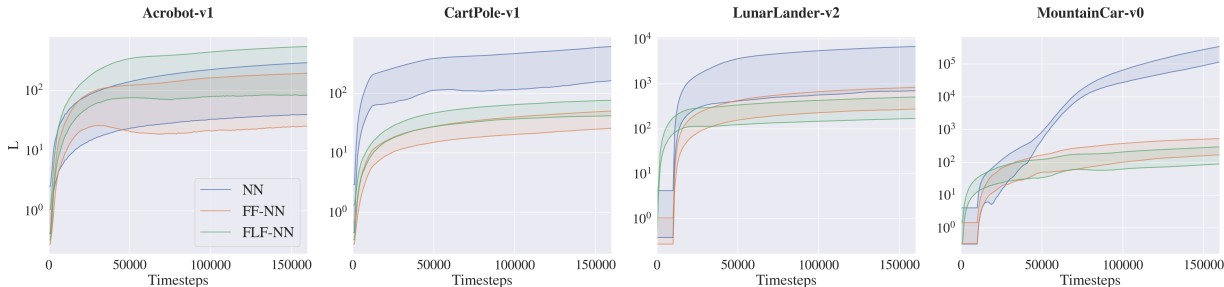

Figure 11: **Preprocessing inputs with Fourier features or Fourier Light features may improve the smoothness of the Neural Network.** Lower and upper bounds on the Lipschitz constant $L$ of NN over training timesteps, for NN fed with raw inputs (blue), FF (orange), and FLF (green). Bounds are averaged over 30 trainings. A lower score is better.

- Average Stiffness (AS): $\mathbb{E}_{(\boldsymbol{s},a),(\boldsymbol{s}',a')\sim\mathcal{B}}[\rho(\boldsymbol{s},a,\boldsymbol{s}',a')]$,

- Average Interference (AI): $\mathbb{E}_{(\boldsymbol{s},a),(\boldsymbol{s}',a')\sim\mathcal{B}}[\rho(\boldsymbol{s},a,\boldsymbol{s}',a')\,|\rho(\boldsymbol{s},a,\boldsymbol{s}',a')<0]$, which only considers (negatively) interfering samples and determines the average of interference,

- Interference Risk (IR): $\mathrm{CVaR}_{0.9}(\rho(\boldsymbol{s},a,\boldsymbol{s}',a')) = \mathbb{E}[\rho(\boldsymbol{s},a,\boldsymbol{s}',a')|\rho(\boldsymbol{s},a,\boldsymbol{s}',a') \leq \overline{\mathrm{VaR}_{0.9}(\rho(\boldsymbol{s},a,\boldsymbol{s}',a'))} \wedge \rho(\boldsymbol{s},a,\boldsymbol{s}',a') \leq 0]$ which is the conditional value at risk of interference where $\mathrm{VaR}_{0.9}(\rho(\boldsymbol{s},a,\boldsymbol{s}',a'))$ is the 0.9-quantile of the distribution of $\rho(\boldsymbol{s},a,\boldsymbol{s}',a')$,

We estimate interference every $1,000$ timesteps on 64 randomly drawn samples from the experience replay buffer and average them over 30 trainings. Our results averaged across all timesteps are reported in Table 2 and curves showing the evolution of interference during the training can be found in Appendix D. In all cases, AS shows that an update with a state-action pair has less impact on other NN predictions with FF/FLF compared to raw inputs. This is confirmed by higher (better) AI and IR scores. Our observations indicate that Fourier features help to generalize appropriately without overgeneralizing and lead to a more stable training and better performance. Interestingly, FLF-NN seems to have even less interference than FF-NN, while we observed sparser representations for FF-NN in Section 4.1. Such results suggest that even if sparsity mitigates catastrophic interference, FLF may have other beneficial effects that reduce catastrophic interference.

Table 3: **Increasing the FLF order improves the metrics**. The table shows Spearman's rank correlation coefficient $r_S$ between different metrics and the order of the FLF. The p-value of the hypothesis test indicates high confidence in the result in almost all cases. The metrics are dead neurons in % (DN), normalized overlap (NO), instance sparsity (IS), Average of Stiffness (AS), Average of Interference (AI), Interference Risk (IR), Lipschitz Lower Bound (LLB), Lipschitz Upper Bound (LUB), averaged across all timesteps for 5 trainings with DQN fed with Fourier Light features (FLF-NN), over an order varying from 1 to 30. ↓ and ↑ indicate the direction in which the metric is better.

| Metric | | MountainCar-v0 | | Acrobot-v1 | | CartPole-v1 | | LunarLander-v2 | |
|---|---|---|---|---|---|---|---|---|---|
| | | $r_S$ | $p$-value | $r_S$ | $p$-value | $r_S$ | $p$-value | $r_S$ | $p$-value |
| DN | ↓ | $-0.876$ | $2.198 \times 10^{-10}$ | $-0.882$ | $1.173 \times 10^{-10}$ | $-0.658$ | $7.768 \times 10^{-5}$ | $-0.869$ | $2.188 \times 10^{-10}$ |
| NO | ↓ | $-0.991$ | $4.529 \times 10^{-26}$ | $-0.93$ | $1.143 \times 10^{-13}$ | $-0.77$ | $6.692 \times 10^{-7}$ | $-0.562$ | $1.009 \times 10^{-3}$ |
| IS | ↓ | $-0.992$ | $2.215 \times 10^{-26}$ | $-0.724$ | $6.082 \times 10^{-6}$ | $-0.75$ | $1.795 \times 10^{-6}$ | $-0.131$ | $4.836 \times 10^{-1}$ |
| $\mathrm{srank}_\delta(\Phi)$ | ↑ | $0.766$ | $8.290 \times 10^{-7}$ | $0.998$ | $7.749 \times 10^{-36}$ | $0.984$ | $2.379 \times 10^{-22}$ | $1.0$ | $0.000$ |
| AS | ↓ | $-0.996$ | $1.256 \times 10^{-31}$ | $-0.955$ | $2.547 \times 10^{-16}$ | $-0.153$ | $4.201 \times 10^{-1}$ | $-0.962$ | $6.575 \times 10^{-18}$ |
| AI | ↑ | $0.994$ | $3.122 \times 10^{-28}$ | $0.968$ | $1.990 \times 10^{-18}$ | $0.501$ | $4.780 \times 10^{-3}$ | $0.942$ | $2.538 \times 10^{-15}$ |
| IR | ↑ | $0.996$ | $1.256 \times 10^{-31}$ | $0.996$ | $6.515 \times 10^{-31}$ | $0.668$ | $5.566 \times 10^{-5}$ | $0.989$ | $1.090 \times 10^{-25}$ |
| LLB | ↓ | $-0.962$ | $2.380 \times 10^{-17}$ | $-0.352$ | $5.631 \times 10^{-2}$ | $-0.828$ | $1.610 \times 10^{-8}$ | $-0.977$ | $5.119 \times 10^{-21}$ |
| LUB | ↓ | $-0.575$ | $8.863 \times 10^{-4}$ | $0.268$ | $1.521 \times 10^{-1}$ | $-0.66$ | $7.228 \times 10^{-5}$ | $-0.98$ | $7.797 \times 10^{-22}$ |

## 4.4 Smoothness of Neural Network

In Neural Networks, larger weights are associated with poorer relative performance on new data (Neyshabur et al., 2015; Bartlett et al., 2017; Neyshabur et al., 2017). The idea follows Occam's razor that models with small weight norms are simpler and perform better than complex models. Not allowing individual weight norms to grow can also discourage large changes in output during the training. Similarly, in Reinforcement Learning, regularization approaches that enforce small weight norms, such as weight decay, tend to produce better results (Farebrother et al., 2018; Liu et al., 2020; Cobbe et al., 2019). A common approach is to normalize weights to ensure that the learned layers are 1-Lipschitz, thereby improving the smoothness of the model, improving convergence (Salimans & Kingma, 2016; Gogianu et al., 2021), and reducing the generalization gap (Rosca et al., 2020; Gouk et al., 2021; Wang et al., 2019). The exact computation of the Lipschitz constant for a NN is NP-hard (Scaman & Virmaux, 2018), but lower bounds and upper bounds can be estimated. The lower bound is obtained by computing for each state-action pair $(\boldsymbol{s}, a)$ in a dataset of $300,000$ state-action pairs the norm of the gradient of the Q-value with respect to the state-action pairs and we report the largest norm encountered (Rosca et al., 2020). Whereas to establish the upper bound, we compute the Lipschitz constants of each layer in isolation and multiply them (Gouk et al., 2021). Under the $l_2$ and $l_1$ norm, the upper bound of the Lipschitz constant of an MLP is given by the spectral norm and the maximum absolute column sum norm measure of the weight matrix (Neyshabur, 2017; Gouk et al., 2021). We measure these bounds every $1,000$ timesteps of the training and we average results over 30 trainings.

In three out of the four tasks shown in Figure 11, NNs with FF features have a lower Lipschitz constant. Additional metrics, based on the $l_1, l_2,$ and $l_\infty$ norm of different layers, indicate that NNs with FLF can lie between NNs with FF and simple NNs, sometimes even surpassing NNs with FF; see Appendix E.

## 4.5 Correlations of the metrics with the FLF order

We finally investigate the correlation between the above metrics and the order of the Fourier Light features. The Spearman's rank correlation coefficients and p-values over an order varying from 1 to 30 in Table 3 indicate that in almost all tasks, increasing the FLF-order is strongly correlated with a better metric and that this correlation is significant. In the CartPole-v1 task, the correlations are weaker, but a closer look at the graphs, shown in Figure 12, suggests that this is due to outliers and saturation in the metric.

When looking at the final reward over varying FLF orders, shown in Fig 13, we observe that increasing the order increases the reward up to a certain point, beyond which performance degrades. Fourier order can be considered as an additional hyperparameter for Deep RL algorithms. Only for the LunarLander-v2 task, the correlation between the reward and the FLF-order is unclear.

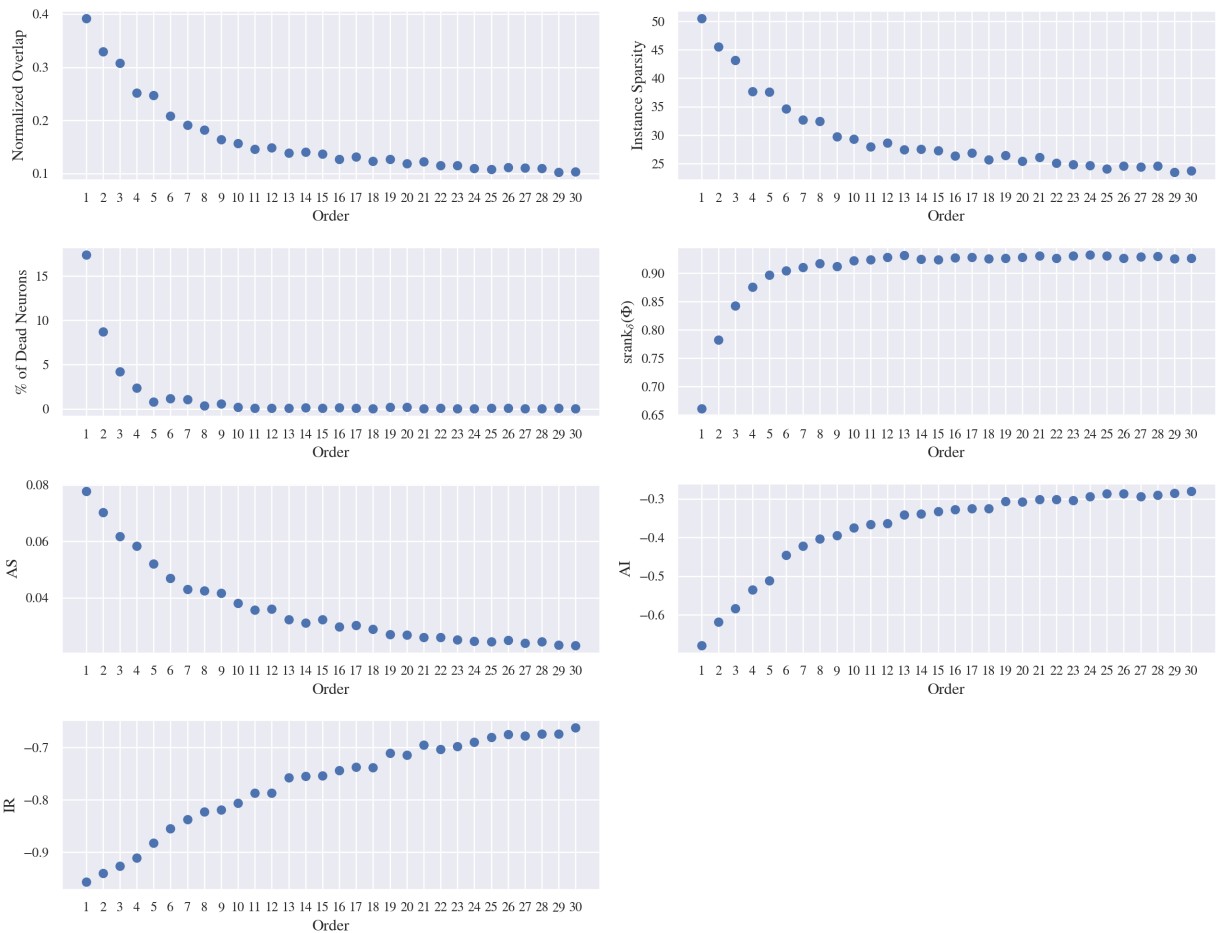

(a) Normalized overlap (NO), instance sparsity (IS), dead neurons in % (DN), srank, Average of Stiffness (AS), Average of Interference (AI), Interference Risk (IR), for CartPole-v1 task

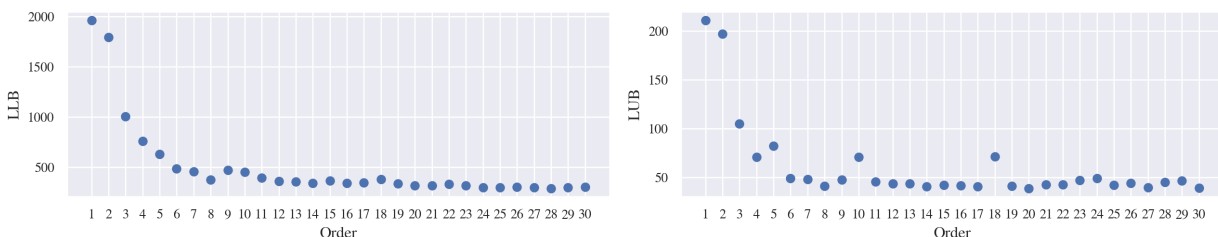

(b) Lipschitz Lower Bound (LLB) and Lipschitz Upper Bound (LUB) for Acrobot-v1 task

Figure 12: Selected metrics over varying FLF orders, for two discrete control tasks

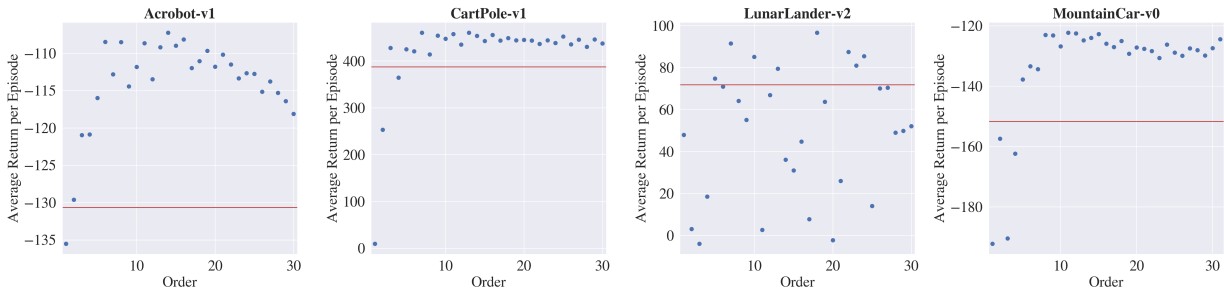

Figure 13: Cumulative rewards over varying FLF orders, averaged across all timesteps for 5 trainings with DQN fed with Fourier Light features. The red line indicates the performance for DQN without any preprocessing.

## 5 Conclusion

We studied the effect of Fourier feature encoding on Deep RL for a set of discrete and continuous control problems. We found that Fourier features provide a systematic increase in the final performance, sample efficiency, learning stability, and robustness to hyperparameters. A detailed empirical analysis of the reason for this behavior showed that Fourier features improve the sparsity, expressiveness, and smoothness of the NN, and reduce catastrophic interference during learning. Additionally, a light version of Fourier features with only a linear number of features compared to the input size leads to similar benefits.

We have just begun to understand how Fourier features and more broadly input preprocessing may improve neural network training. Our work suggests thus several promising future research directions. The first is to investigate whether Fourier features behave as an implicit regularizer and decrease the generalization gap. Second, we plan to further study the correlations between performance, sparsity, expressiveness, learning interference, and smoothness of neural networks.

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

## A  Additonal Experiment on Ant

This section shows the following additional empirical result on a larger scale domain which cannot be put in the main body due to space limitations. For this experiment, we apply only FLF to PPO and compare performance to PPO without encoding We only test FLF because the number of standard Fourier Features explodes due to the higher state dimension. PPO is applied to Neural Network with the ELu activation function. For computation time reasons and fair comparison, only the learning rate and Fourier order are re-optimized for FLF-NN. Other hyperparameters are taken from Nivida Isaac Gym Environments (Makoviychuk et al., 2021). Figure 14 shows the averaged returns per episode of PPO on Ant continuous-action control task incorporating 60 kinematic observations from Nivida Isaac Gym Environments (Makoviychuk et al., 2021). As in Section 3, FLF have better performance in terms of cumulative rewards with fewer environment interactions.

## B  Comparison With Other Input Preprocessing

FF/FLF provide clear benefits, but it is natural to ask if other simple input preprocessing could provide similar advantages. In this section, we propose to empirically compare the performance of Neural Networks fed with Fourier features and with other traditional hand-designed features used in standard RL. We first give a brief description of the different preprocessing we investigate in our experiments and then we perform a comparison on MountainCar-v0 and CartPole-v1. We compare Fourier features with three standard input preprocessings:

- **Polynomial Features** (PF-NN). Polynomials are one of the simplest families of features used for interpolation and regression. The feature vector consists of all polynomial combinations of the state variables with a degree less than or equal to a specified degree (Lagoudakis & Parr, 2003; Sutton & Barto, 2018).

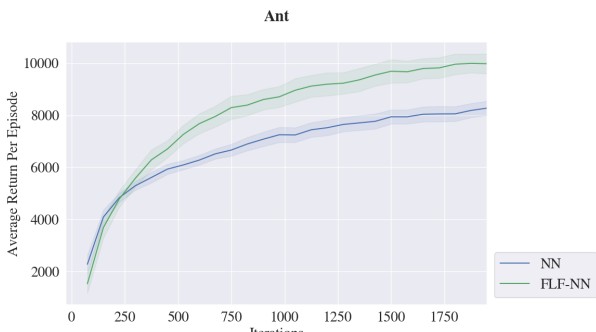

Figure 14: **The use of Fourier Light features improves the performance and sample efficiency of PPO on Ant.** Evaluation learning curves of NN (blue) and FLF-NN (green), reporting episodic return versus environment timesteps. Results are averaged over 30 training with shading indicating the 95% confidence interval (CI).

- **Random Fourier Features (RFF)**. Random Fourier Features were initially introduced to approximate an arbitrary stationary kernel-invariant by exploiting Bochner's theorem (Rahimi & Recht, 2007). Recent works have shown promising results where RFFs boost the performance of deep neural networks (Mehrkanoon & Suykens, 2018), reduce the probability of misclassification (Mitra & Kaddoum, 2021), or can facilitate MLPs to learn high-frequency functions (Tancik et al., 2020). In RL, Rajeswaran et al. (2017) used them with Natural Policy Gradient to outperform performance obtained with NNs. The $i$-th feature of the Random Fourier Feature mapping $\text{RFF} : \mathbb{R}^n \to \mathbb{R}^p$ is

$$\text{RFF}_i(\boldsymbol{s}) = {}^2\!/\!\sqrt{p} \cos\left(\boldsymbol{s}^T \boldsymbol{c}_i + b_i\right) \tag{7}$$

  where $\boldsymbol{c} \sim \mathcal{N}(\boldsymbol{0}, \sigma^2 \boldsymbol{I}_p)$, $\boldsymbol{b} \sim \mathcal{U}(0, 2\pi)$, and $p$ is the number of features we want to generate. The term $2/\sqrt{p}$ is used as a normalization factor to reduce the variance of the estimates. RFFs and Fourier features have a very close definition, except that the vector $\boldsymbol{c}$ creating interaction between state variables is sampled from a normal distribution. RFFs are studied to understand if it is rather the structure or the choice of $\boldsymbol{c}$ that can explain the good performances.

- **Tile Coding (TC)**. Tile Coding (Albus, 1971; Sutton & Barto, 2018) is a generalization of state aggregation, in which we cover the state space with overlapping grids (tilings) where each grid divides the state space into small squares (tiles). The representation of a state for each tiling is a one-hot vector of dimension the number of tiles that has one for the tile where the state is in and zero otherwise. Concatenation of one-hot vectors for each tiling forms Tile Coding features. A nice property of Tile Coding is that generalization occurs to states other than the one trained if those states fall within any of the same tiles. Ghiassian et al. (2020) already proposed to preprocess inputs of Neural Network with Tile Coding to promote sparsity of learned representations and obtain better performance.

In the experiments shown in Figure 15, the ranking on final average rewards is: PF-NN < RFF-NN < NN < TC-NN < FF-NN ≃ FLF-NN where < means lower performance. None of the other input preprocessings achieve the performance of FF-NN/FLF-NN, even though we tuned their hyperparameters through an extensive search. It is even worse since PF and RFF degrade performance. TC-NN does better than NN for the MountainCar-v0 but has slightly the same performance for CartPole-v1. These results suggest that applying input preprocessing to NN does not necessarily improve performance and can even degrade them. In the following, we evaluate the effects on the Neural Network using the metrics from Section 4. **Sparsity** measures, as reported in Table 4, suggest that standard input preprocessings degrade sparsity. Even Tile Coding, reported to promote sparsity in (Ghiassian et al., 2020), produces less sparse representations than standard DQN. Tile Coding, just as FF/FLF, does not have dead neurons, while PF-NN and RFF-NN increase the number of dead neurons.

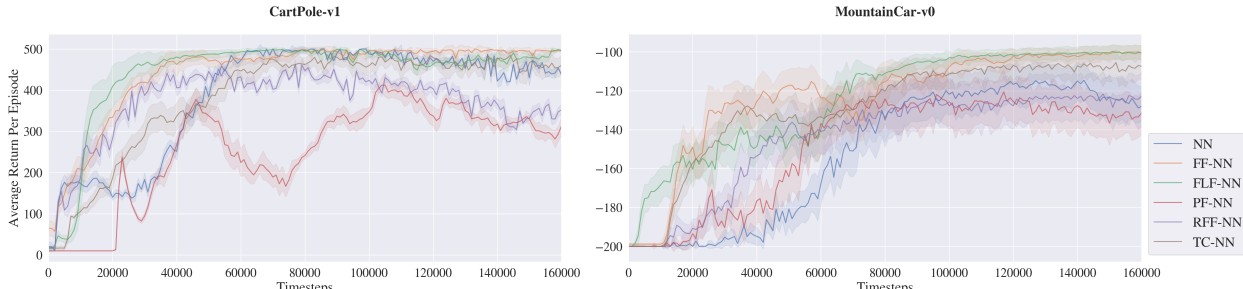

Figure 15: **Fourier Features/Fourier Light features outperform other standard input preprocessings on discrete control tasks.** Evaluation learning curves of NN (blue), FF-NN (orange), FLF-NN (green), PF-NN (red), RFF-NN (purple) and TC-NN (brown) reporting episodic return versus environment timesteps. Results are averaged over 30 trainings with shading indicating the standard deviation.

Table 4: **Input preprocessings do not necessarily promote sparsity in discrete control tasks**. Sparsity scores with dead neurons in % (DN), normalized overlap (NO) and instance sparsity (IS) obtained for DQN fed with raw inputs (NN), Fourier features (FF-NN), Fourier Light features (FLF-NN), Normalized inputs (NI-NN), Polynomial features (PF-NN), Random Fourier features (RFF-NN) and Tile Coding (TC-NN) on discrete control tasks averaged across all timesteps. Averages and margins of error of the 95% CI are over 30 trainings. Lower scores better.

| Task | | NN | FF-NN | FLF-NN | PF-NN | RFF-NN | TC-NN |
|---|---|---|---|---|---|---|---|
| **MountainCar-v0** | DN | $0.47 \pm 0.09$ | **0.0** | **0.0** | $0.66 \pm 0.08$ | $0.48 \pm 0.04$ | **0.0** |
| | NO | $0.72 \pm 0.08$ | **$0.37 \pm 0.06$** | $0.43 \pm 0.10$ | $0.80 \pm 0.08$ | $0.87 \pm 0.06$ | $0.77 \pm 0.13$ |
| | IS | $0.78 \pm 0.07$ | **$0.57 \pm 0.05$** | $0.62 \pm 0.08$ | $0.84 \pm 0.08$ | $0.90 \pm 0.05$ | $0.86 \pm 0.09$ |
| **CartPole-v1** | DN | $0.07 \pm 0.02$ | $0.01$ | **0.0** | $0.23 \pm 0.02$ | $0.88 \pm 0.07$ | **0.0** |
| | NO | $0.63 \pm 0.04$ | **$0.52 \pm 0.02$** | $0.79 \pm 0.07$ | $0.73 \pm 0.07$ | $0.58 \pm 0.03$ | $0.66 \pm 0.06$ |
| | IS | $0.66 \pm 0.03$ | **$0.60 \pm 0.02$** | $0.85 \pm 0.06$ | $0.75 \pm 0.05$ | **$0.61 \pm 0.03$** | $0.70 \pm 0.04$ |

**Expressiveness** is indicated by the learning curves of effective rank in Figure 16. PF-NN and RFF-NN generate poorer features than NN fed with raw inputs. As expected given the absence of dead neurons, TC-NN produces richer features than NN and is on par with FF/FLF-NN.

**Interference** scores, shown in Table 5, indicate that PF-NN, RFF-NN, and TC-NN highly interfere during the training with poor Average of Interference (AI) and Interference Risk (IR) scores. These results are consistent with the poor sparsity scores.

**Smoothness**, as measured by the Lipschitz bounds shown in Figure 17, is improved by all input preprocessings, with TC/FF/FLF giving the best performance, followed by PF/RFF. In the shown instance, smoother networks correlate with better learning performance.

## C    Sparsity Learning Curves for DQN on Discrete Control Tasks

Normalized overlap as function of environment steps corresponding is reported in Figure 18. The average of these scores across timesteps can be found in Table 1. We estimate sparsity scores every $1,000$ timesteps and average them over 30 trainings.

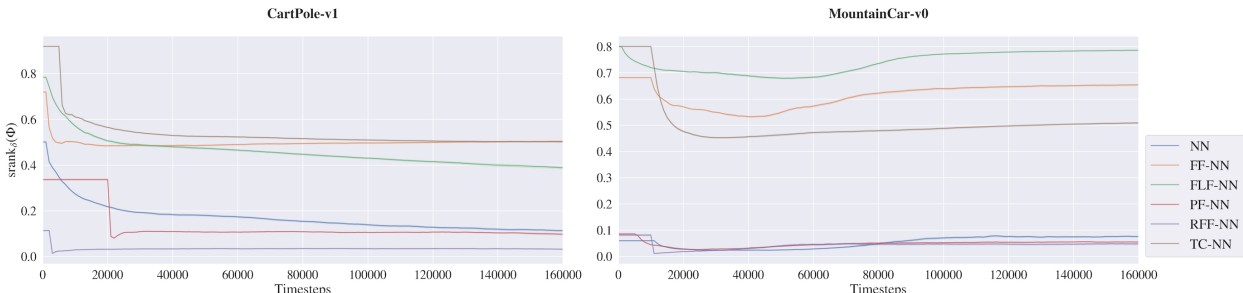

Figure 16: Learning curves of the normalized effective rank $\mathrm{srank}_\delta(\mathbf{\Phi})$ for NN fed with raw inputs (blue), Fourier features (orange), Fourier Light features (green), Polynomial features (red), Random Fourier Features (purple) and Tile Coding features (brown) on discrete control tasks. Results are averaged over 30 training with the shade that indicates the 95% CI.

Table 5: Interference measures with Average of Stiffness (AS), Average of Interference (AI), Interference Risk (IR), Percentage of Interference within a batch (PercI) and Average Q-Interference (AQI) averaged across all timesteps for standard input preprocessings on discrete control tasks where ↓ and ↑ mean lower and higher the score is better, respectively.

| Tasks | | | NN | FF-NN | FLF-NN | PF-NN | RFF-NN | TC-NN |
|---|---|---|---|---|---|---|---|---|
| **MountainCar-v0** | AS | ↓ | 0.24 | 0.1 | **0.05** | 0.21 | 0.14 | 0.1 |
| | AI | ↑ | −0.83 | **−0.47** | −0.54 | −0.81 | −0.88 | −0.86 |
| | IR | ↑ | −0.91 | **−0.87** | **−0.87** | −0.95 | −0.93 | −0.93 |
| | PercI | ↓ | **0.38** | 0.44 | 0.47 | 0.39 | 0.43 | 0.44 |
| **CartPole-v1** | AS | ↓ | 0.22 | **0.05** | **0.05** | 0.13 | 0.49 | **0.07** |
| | AI | ↑ | −0.92 | **−0.73** | −0.86 | −0.84 | −0.97 | −0.95 |
| | IR | ↑ | −0.99 | −0.98 | −0.98 | **−0.87** | −0.99 | −0.97 |
| | PercI | ↓ | 0.39 | 0.47 | 0.47 | 0.43 | **0.25** | 0.46 |

# D  Interference Learning Curves for DQN on Discrete Control Tasks

**Evolution of Interference During the Training.**  We estimate interference every $1,000$ timesteps and average them over 30 trainings. For each point in the learning curves, we randomly draw 64 samples from the experience replay buffer to estimate the interference. All interference measures are defined in Section 4.3.

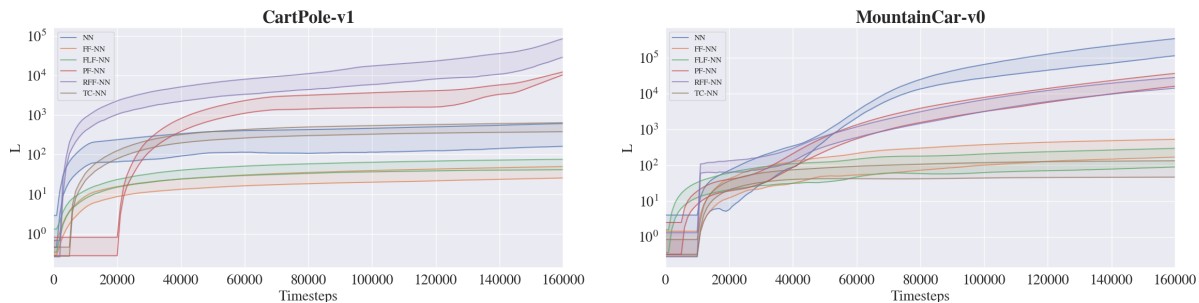

Figure 17: Lower and upper bounds on the Lipschitz constant of NN over training timesteps, of NN (blue), FF-NN (orange), FLF-NN (green), PF-NN (red), RFF-NN (purple), and TC-NN (brown). Bounds are averaged over 30 trainings with shading indicating the 95% CI

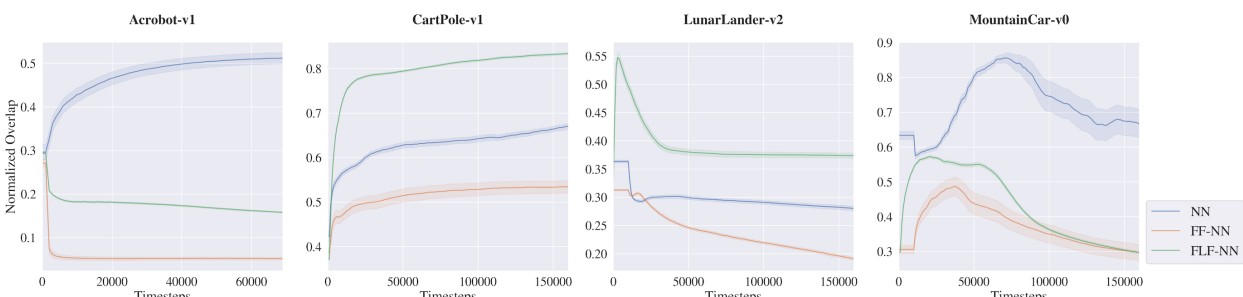

Figure 18: Learning curves of the normalized overlap for NN (blue), FF-NN (orange), and FLF-NN (green) with DQN on discrete control tasks, reporting normalized overlap versus environment timesteps studied in Table 1. Results are averaged over 30 trainings (different seeds), with shading indicating the 95% CI.

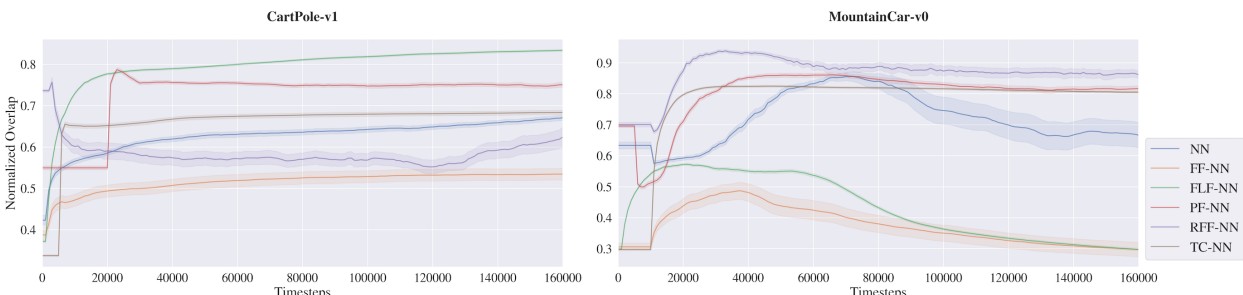

Figure 19: Learning curves of the normalized overlap for NN (blue), FF-NN (orange), FLF-NN (green), PF-NN (red), RFF-NN (purple), and TC-NN (brown) with DQN on discrete control tasks, reporting normalized overlap versus environment timesteps studied in Table 4. Results are averaged over 30 trainings (different seeds), with shading indicating the 95% CI.

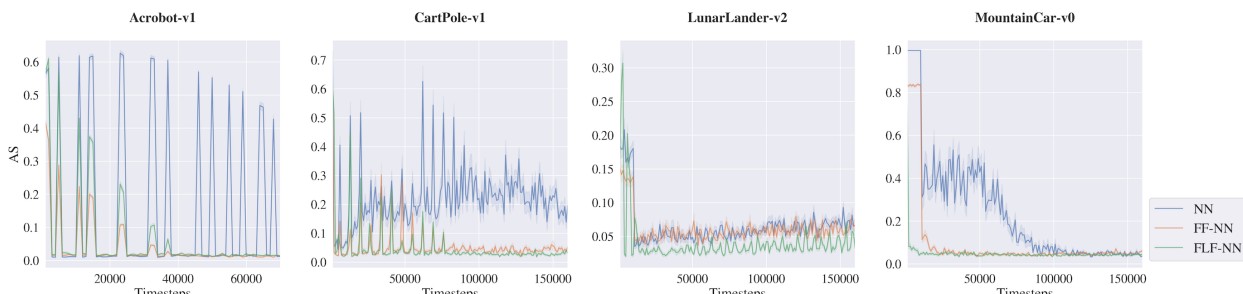

Figure 20: Learning curves of the **Average Stiffness** (AS) for NN fed with raw inputs (blue), Fourier features (orange), and Fourier Light features (green) studied in Table 2. Results are averaged over 30 trainings with shading indicating the 95% CI.

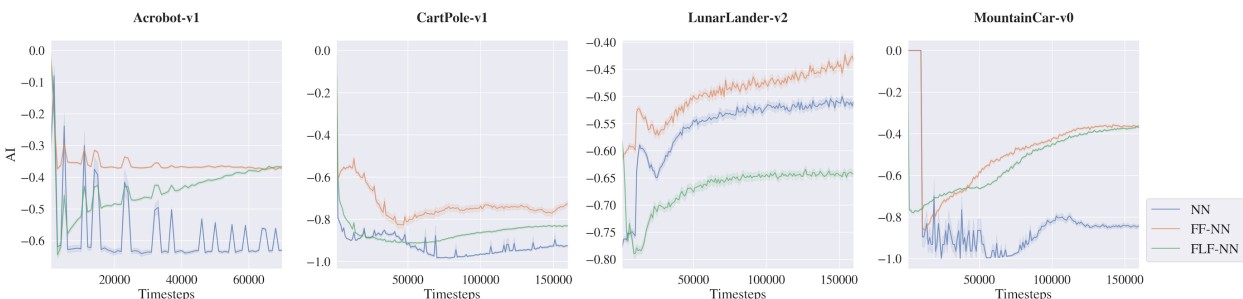

Figure 21: Learning curves of the **Average Interference** (AI) for NN fed with raw inputs (blue), Fourier features (orange), and Fourier Light features (green) studied in Table 2. Results are averaged over 30 trainings with shading indicating the 95% CI.

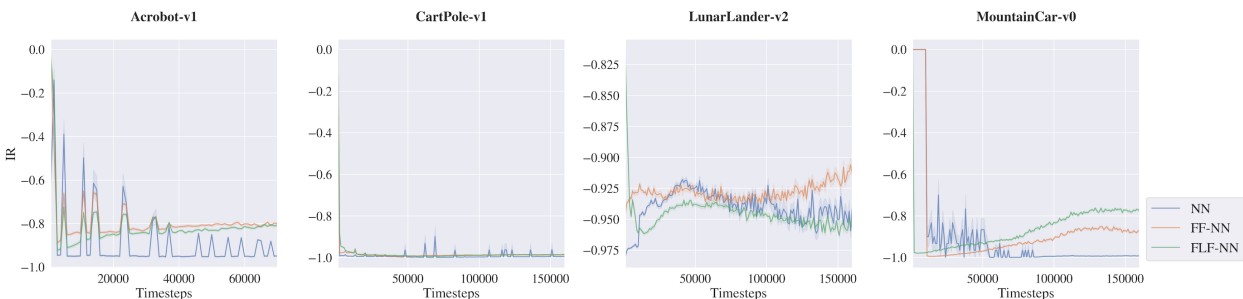

Figure 22: Learning curves of the **Interference Risk** (IR) for NN fed with raw inputs (blue), Fourier features (orange), and Fourier Light features (green) studied in Table 2. Results are averaged over 30 trainings with shading indicating the 95% CI.

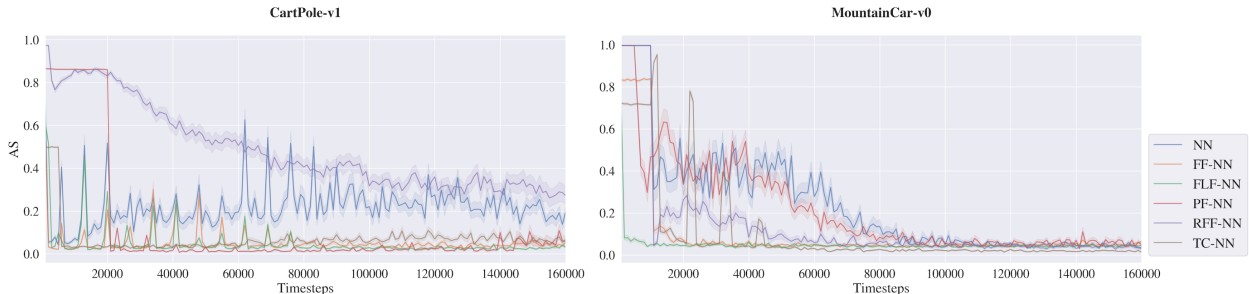

Figure 23: Learning curves of the **Average Stiffness** (AS) for NN fed with raw inputs (blue), Fourier features (orange), Fourier Light features (green), PF-NN (red), RFF-NN (purple), and TC-NN (brown) studied in Table 5. Results are averaged over 30 trainings with shading indicating the 95% CI.

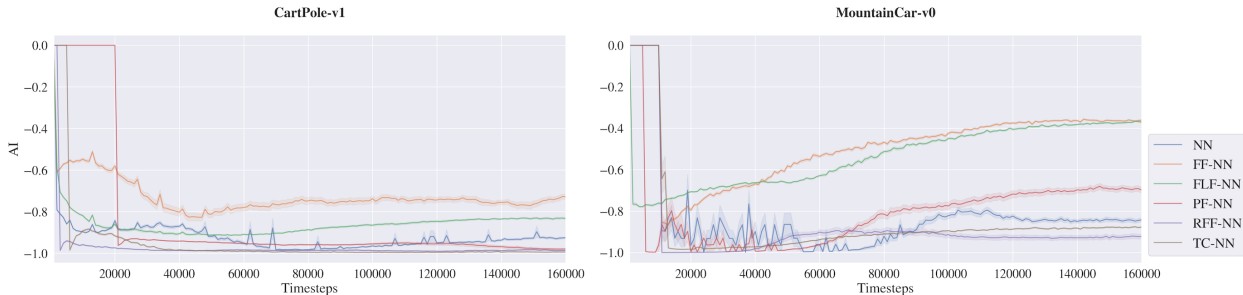

Figure 24: Learning curves of the **Average Interference** (AI) for NN fed with raw inputs (blue), Fourier features (orange), Fourier Light features (green), PF-NN (red), RFF-NN (purple), and TC-NN (brown) studied in Table 5. Results are averaged over 30 trainings with shading indicating the 95% CI.

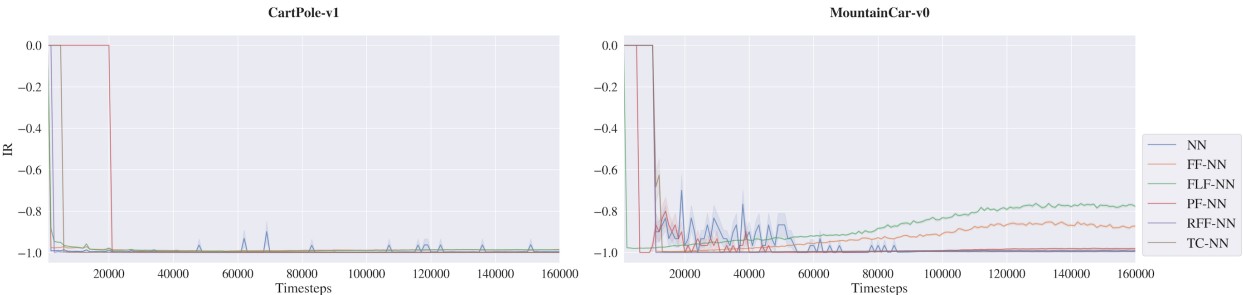

Figure 25: Learning curves of the **Interference Risk** (IR) for NN fed with raw inputs (blue), Fourier features (orange), Fourier Light features (green), PF-NN (red), RFF-NN (purple), and TC-NN (brown) studied in Table 5. Results are averaged over 30 trainings with shading indicating the 95% CI.

# E  Smoothness of Neural Networks for DQN on Discrete Control Tasks

**Estimation of the Smoothness of the Neural Network.**  The exact computation of the Lipschitz constant for a NN is NP-hard (Scaman & Virmaux, 2018), but lower bounds and upper bounds can be estimated. The lower bound is obtained by computing for each state-action pair $(s, a)$ in a dataset $300,000$ state-action pairs the norm of the gradient of the Q-value with respect to the state-action pairs and we report the largest norm encountered (Rosca et al., 2020). Whereas to establish the upper bound, we compute the Lipschitz constants of each layer in isolation and multiply them (Gouk et al., 2021). Under the $l_2$ and $l_1$ norm, the upper bound of the Lipschitz constant of an MLP is given by the spectral norm and the maximum absolute column sum norm measure of the weight matrix (Neyshabur, 2017; Gouk et al., 2021). We measure these bounds every $1,000$ timesteps of the training and we average results over 30 trainings.

**Simplicity of the Neural Network.**  Liu et al. (2020) observed that on many tasks, smaller policy weight norms correlate with better generalization ability. Figure 26, Figure 26, Figure 28 visualize the $l_2$, $l_1$, $l_{\text{inf}}$ weight norms on both layers, respectively. In most cases, FF and FLF reduce the $l_2$ and $l_1$ weight norms, especially in the second layer.

# F  Reproducing Experiments

The code for reproducing all experiments is available on GitHub at `https://github.com/DavidBrellmann/Fourier_Features_in_RL_with_NN`.

**Discrete Control Tasks.**  We compare the performance of DQN on five discrete-action environments from OpenAI Gym (Brockman et al., 2016): Acrobot-v1, CartPole-v1, LunarLander-v2, MountainCar-v0, and Catcher-v1 (Tasfi, 2016). We use an MLP architecture with a single hidden layer.

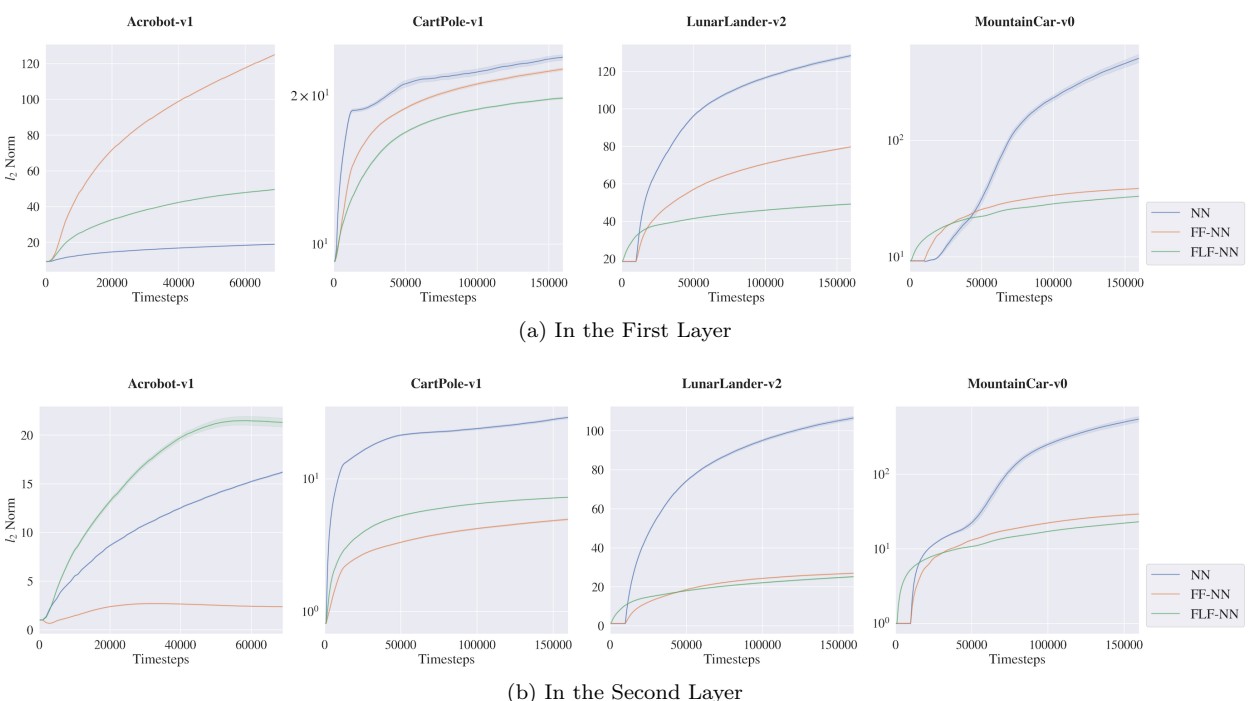

(a) In the First Layer

(b) In the Second Layer

Figure 26: Learning curves of the $l_2$ weight norm for NN fed with raw inputs (blue), Fourier features (orange), and Fourier Light features (green), averaged over 30 trainings with shading indicating the 95% CI.

**Continuous Control Tasks.** We compare performance on three continuous-action control tasks from Mujoco (Todorov et al., 2012): HalfCheetah-v2, InvertedDoublePendulum-v2, and Swimmer-v2. Because of the higher state dimension, we were only able to test Fourier Light Features. We use a MLP with two hidden layers. All experiments are averaged over 10 runs, with offline evaluations performed on the policy every 1,000 environment timesteps.

**Common Settings.** The first for discrete action domains are from OpenAI Gym (Brockman et al., 2016) with version 0.18.0 and continuous action domains from Mujoco (Todorov et al., 2012). The discrete action environment Catcher-v1 is from PyGame-Learning-Environment (Tasfi, 2016). For Deep Reinforcement Learning implementations, we adopt the code from StableBaselines-3 (Raffin et al., 2019) with version 0.10.0 based on Pytorch 1.8.0. We use Adam optimizer (Kingma & Ba, 2014), Xavier initializer (Glorot & Bengio, 2010), and ReLU activation functions across all experiments. We evaluate offline each algorithm every 1,000 training/environment timesteps.

**Hyperparameter Sampling on Discrete Control Tasks.** Since hyperparameters for the discrete control tasks were not included at the time of experiments, we tune hyperparameters with Optuna 2.4.0 (Akiba et al., 2019) for each discrete control task. We sample hyperparameter values in Table 6 using the TPE (Tree-structured Parzen Estimator) algorithm (Bergstra et al., 2011). Since TPE can be very sensitive to the scores of the first trials, we run 5 independent hyperparameter research where one research is composed of 500 trials. Each trial corresponds to a training of 120,000 timesteps with hyperparameters sampled by TPE, and the score of each trial is based on the return of 100 rollouts of the learned policy. Because Deep RL algorithms are highly unreliable (Henderson et al., 2018; Islam et al., 2017), we take the 15 best hyperparameter settings returned by Optuna and run 5 additional trainings of 150,000 timesteps. The best hyperparameter setting we take is the one with the better final average return over the 5 trainings.

For searching the learning rate and Fourier Order of FF and FLF, we adopt the same strategy by doing just one research with Optuna. For the Fourier Light order, we sample values between 1 and the Fourier Light order corresponding to the number of traditional Fourier features found previously for the research for FF.

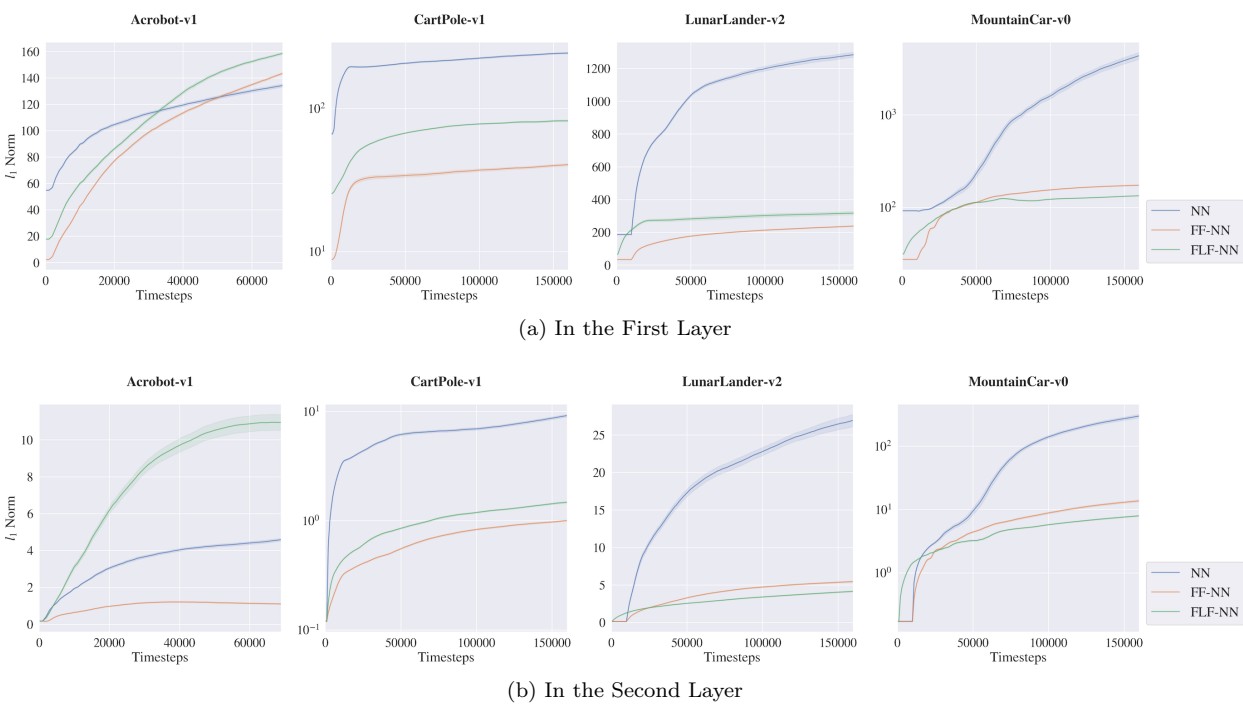

Figure 27: Learning curves of the $l_1$ weight norm for NN fed with raw inputs (blue), Fourier features (orange), and Fourier Light features (green), averaged over 30 trainings with shading indicating the 95% CI.

Table 6: **Sampling Values for DQN**

| Hyperparameter | Range |
| --- | :---: |
| Number of Hidden Layers | 1 |
| Number of Neurons per Hidden Layer | $\{16, 32, 64, 128, 256\}^1$ |
| Batch Size | $\{16, 32, 64, 100, 128, 256, 512\}$ |
| Replay Buffer Size | $\{1e4, 5e4, 1e5, 1e6\}$ |
| Discount Factor | $\{0.9, 0.95, 0.98, 0.99, 0.995, 0.999, 0.9999\}$ |
| Learning rate | $[1e-5, 1]$ |
| Target Update Frequency | $\{0.9, 0.95, 0.98, 0.99, 0.995, 0.999, 0.9999\}$ |
| Train Frequency | $\{1, 4, 8, 16, 128, 256, 1000\}$ |
| Exploration Fraction | $[0, 0.5]$ |
| Final Value of Random Action Probability | $[0, 0.2]$ |
| Fourier Order (FF) | $\{1, 2, 3, 4, 5\}$ |

**Hyperparameter Sampling on Continuous Control Tasks.** On continuous tasks, we keep the hyperparameters from the original codebase, StableBaselines-3-zoo (Raffin, 2020). For searching the learning rate and Fourier Light Order in continuous control tasks, we adopt the same strategy as the one detailed in the previous subsection by doing just one research of 500 trials with Optuna. In this setting, each trial corresponds to a training of $800,000$ timesteps with hyperparameters sampled by TPE. Similarly, the best hyperparameter setting we take is the one with the better final average return over 5 trainings of $800,000$ timesteps. The range of values for the Fourier Light order in continuous control tasks is between 1 and 50.

**Normalization.** Before being passed into the Fourier features mapping or Fourier Light features mapping, input data need to be normalized by a normalizer $\psi : \mathbb{R}^n \to [0,1]^n$ (Section 3). We compute either the min/max normalization when state variables are bounded or we apply a nonlinear transformation based on

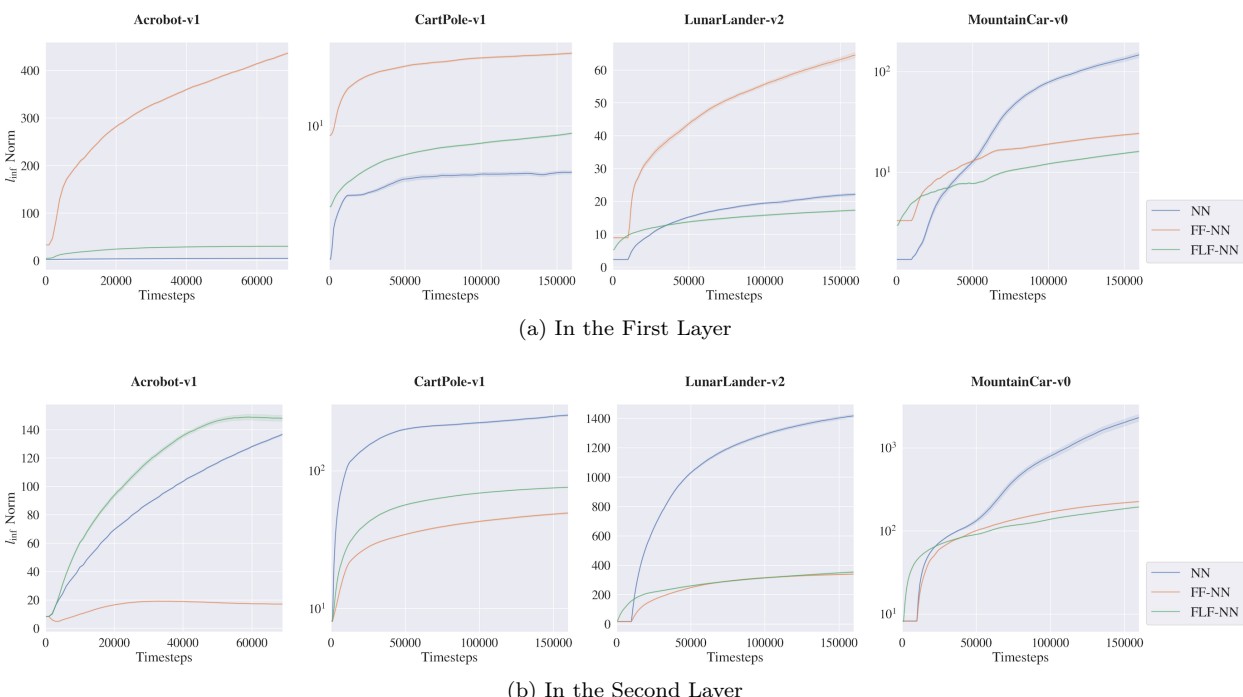

Figure 28: Learning curves of the $l_{\inf}$ weight norm for NN fed with raw inputs (blue), Fourier features (orange), and Fourier Light features (green), averaged over 30 trainings with shading indicating the 95% CI.

arctan where:

$$\psi_i(\boldsymbol{s}) = \frac{\arctan(\frac{\boldsymbol{s}_i - \boldsymbol{t}_i}{\boldsymbol{c}_i})}{\pi} + 0.5 \tag{8}$$

with $\boldsymbol{t}$ the normalizer shift parameter and $\boldsymbol{c}$ the normalizer scale parameter. $\boldsymbol{t}$ and $\boldsymbol{c}$ are arbitrarily chosen to obtain $\psi_i(\boldsymbol{s}_{\max}) = 0.9$ and $\psi_i(\boldsymbol{s}_{\min}) = 0.1$ where $\boldsymbol{s}_{\max,i}$ and $\boldsymbol{s}_{\min,i}$ are respectively the maximum and minimum state variable $i$ values observed over $5 \times 5,000$ rollouts generated by 5 suboptimal policies.

**Training timesteps.** For DQN, we run $160,000$ timesteps for Acrobot-v1, CartPole-v1, Catcher-v1, LunarLander-v1 and MountainCar-v0. For Catcher-v1, we fix the episode length to 500 timesteps. For PPO, we run $2e8$ timesteps for HalfCheetah-v2 and Swimmer-v2 and $1e8$ timesteps for InvertedDoublePendulum-v2.

**Implementation Details for Standard Input Preprocessing.** For generating Polynomial features we use Scikit-Learn (Buitinck et al., 2013) and for Tile Coding features we adopt the implementation of Patterson (2020). For Random Fourier features and Tile Coding features, inputs are first normalized with the same normalization defined for Fourier features/Fourier Light features. In our first experiments, we noticed that just changing the learning rate was not sufficient to achieve convergence. We then keep the same MLP architecture and found another hyperparameter setting with Optuna by using the same hyperparameter ranges reported in Table 6. According to the experiments performed by Ghiassian et al. (2020), we set the number of tilings to 8 and the number of tiles to 4.

