# OpenReview forum: "Fourier Features in Reinforcement Learning with Neural Networks"
_TMLR — Accepted by TMLR_

### Review · Reviewer_Yrod · 2023-04-19

**Summary Of Contributions:**

This paper has two core contributions:
- a low-rank version of Fourier features (FF), Fourier Light Features (FLF)
- a thorough investigation of the use of Fourier features in (small scale) deep RL

More specifically, the proposed FLF assume that input dimensions are in some sense independent in the frequency domain. The intuition is that this provides a better basis for a non-linear model (like an MLP) to then decide how input dimensions are interrelated. This means that instead of $O(m^n)$ features we have $O(mn)$ features, for $m$ the number of frequency components, $n$ the input dimensionality.

The investigation reveals that FF & FLF:
- can more easily model high-frequency value functions
- increase sparsity in MLP representations
- increase the effective rank (and thus presumably the expressivity) of representations
- sometimes have some beneficial effects on interference
- learn functions with a lower (approximate) Lipschitz constant, i.e. smoother functions
- induces an intuitive capacity knob, $m$, which is predictive of improved performance (and of the above metrics)


**Audience:**

Yes

**Broader Impact Concerns:**

No specific concerns.

**Claims And Evidence:**

Yes

**Requested Changes:**

Some comments:
- "state-based RL", do the authors mean "observation based RL"? I'm not sure what "state" refers to here, presumably all RL is based on states?
- it would be nice to have consistent font sizes on figures (ideally figure legends are at most 2 pts below the main text's)
- Using $n(m+1)$ input features instead of $n$ input features in an MLP adds an extra $mn$ learnable parameters. This is non-negligible. I don't see an experiment that accounted for this discrepancy between FNF and NN (nor FF actually), unless I've misunderstood the setup.

**Strengths And Weaknesses:**

The paper is well written, it felt easy to read, although I was already quite familiar with the cited papers, such as Dong et al., Lyle et al., or Kumar et al.'s works. I wonder if readers less familiar with this literature might benefit from a bit more hand holding.

The work in the paper is good science, it builds upon past investigations to thoroughly examine a proposed method and gain insights into its benefits.

The main weakness I can see is the scale of the tackled problems; I appreciate results at this scale, and I don't see it as a barrier to acceptance, but I think applying FLF to one larger scale domain would be beneficial. In particular, as the authors point out, a sufficiently large DNN eventually can model any function. Orthogonally to image domains like Atari or ProcGen (where convolutions are already Fourier-like), it may be a valuable contribution to identify the _types_ of (larger scale) domains where Fourier features are particularly important.

---

> ### Author Response · Authors · 2023-05-15
>
> Dear Reviewer Yrod,
>
> Thank you for your time and effort in reviewing our work. We have provided detailed clarification to address the issues raised in your comments. For your convenience, all significant changes have been highlighted in red text within the paper.
>
> Thank you for pointing out the inaccuracy of the term "state-based environment". We have now opted for the term "kinematic observation-based" to describe our environments more precisely. We would like to emphasize that in this paper, our focus is exclusively on MLPs that operate with kinematic observations. For a study on the impact of Fourier series on convolutional layers, we refer you to the concurrent work by Benbarka, N., Höfer, T., & Zell, A. (2022) titled "Seeing implicit neural representations as Fourier series" published in the Proceedings of the IEEE/CVF Winter Conference on Applications of Computer Vision. We have included an additional experiment on Ant with PPO—a large-scale domain featuring 60 kinematic observations.
>
> In response to your comment regarding consistency, we have also revised the figures in our paper to maintain uniform font sizes.
>
> It is true that the expansion of input parameters with FF and FLF leads to an increase in learnable parameters. However, as underscored in Section 2, it is not conclusive that this increase contributes to the improved performance of FF and FLF. Even with overparameterized neural networks, NNs still fail to learn value functions.
>
> If you have any additional questions or comments, we would be happy to have further discussions.
>
> Kind regards,
>
> The authors.

---

### Review · Reviewer_mfm9 · 2023-04-21

**Summary Of Contributions:**

This paper discusses how encoding inputs with a Fourier feature mapping can improve the performance of deep reinforcement learning models. Experiments show that it can lead to performance gains and increased robustness with respect to hyperparameters. The authors propose a specific version of Fourrier features that only has a linear number of features. The experiments cover a variety of RL settings.

**Audience:**

Yes

**Claims And Evidence:**

No

**Requested Changes:**

- The toy environment from Figure 2 is not very clearly described. First, the description of the MDP does not seem to be recalled in the paper but the reader is sent to another paper (would it be possible to describe it, at least in one or two sentences?).Then, for this toy environment, what would the Fourrier features look like given that the state is apparently only one scalar between 0 and 1? I do not understand the discussion in the last paragraph of Section 2 that discusses the idea of "high-frequency" components that are important, etc.
- From Figure 4, it is unclear whether the blue (baseline) is significantly worse in 3 out of 4 of the experiments. Can it be clarified?
- It would be interesting to provide a visualization of a few states and the corresponding Fourrier features for some of the environments to provide a better idea about what is fed to the the network in at least some of the cases. It would also clarify whether the pixels are used for the states or whether it is the low-dimensional state. It seems that only a few discrete features are provided (e.g. see Before Equation 2)? Does that mean for instance that for mountain car, only two features are used?
- Some other important technical elements are not clearly mentioned, or at least not in the main text, which makes the reading not as informative as it could be. For instance, the ReLU activation functions are only mentioned in the appendix while it is a key element to discuss the "dead neurons" with zero activation.
- FQI is mentioned then DQN is but not much detail are provided at least for FQI.
- Figure 22, 23 (among others have a standard deviation that is zero for some of the curves? Is that normal?
- (minor) Sometimes the word "Figure" is used to refer to a figure, sometime "Fig".

**Strengths And Weaknesses:**

Strength:
- The results provided are quite convincing with respect to the claim that Fourrier features can be beneficial.

Weaknesses:
- Some technical elements are not described with all the necessary details. As it is currently, key elements are unclear to me (see below).
- (minor) The authors mention that the code will be made available. Could it be shared at this point such that reviewers can check it?

---

> ### Author Response · Authors · 2023-05-15
>
> Dear Reviewer mfm9,
>
> Thank you for your time and effort in reviewing our work. We have provided detailed clarification to address the issues raised in your comments and added new figures with visualization of Fourier Features and Fourier Light Features. For your convenience, we have highlighted these significant modifications in red text.
>
> Thank you for pointing out the inaccuracy of the term "state-based environment". We have now opted for the term "kinematic observation-based" to describe our environments more precisely. Please note that in this paper, we have chosen to concentrate only on MLPs that are fed with kinematic observations.
>
> Regarding the code, we are currently in the process of refining and organizing it to ensure clarity and ease of use. It will be made available online as soon as possible.
>
> We hope that our revisions and clarifications have addressed your concerns. We would be grateful if you could re-evaluate our work.
>
> If you have any additional questions or comments, we would be happy to have further discussions.
>
> Kind regards,
>
> The authors

---

### Review · Reviewer_1EW8 · 2023-05-02

**Summary Of Contributions:**

This paper empirically investigates the use of fixed Fourier Feature representations as a preprocessing step for neural network function approximation in reinforcement learning. The paper investigates the DQN and PPO learning systems equipped with two different preprocessing steps alongside their standard implementation and show across several domains that the fourier feature preprocessing step significantly improves performance, increases smoothness of the representation (measured by Lipschitz constant), increases the stable rank of the weight matrices, reduces the number of dead neurons, and decreases sensitivity to hyperparameters (stepsize, buffer size, target net refresh rate).

**Audience:**

Yes

**Broader Impact Concerns:**

No broader impact concerns.

**Claims And Evidence:**

Yes

**Requested Changes:**

* Bottom of page 2: It isn't clear to me that the claim "Such complexities arise from the recursive applications of the Bellman operator." is true. Because we typically use discounting, recursively applying the Bellman operator is a filtration process---typically smoothening out the values. The lack of smoothness here likely is coming from the Bellman optimality operator, which has the maximization over actions which can cause high complexity. Anyways, this is a trivial wording nitpick.

* Figure 3 would benefit from a different statistic than standard deviation. It isn't clear how shaded regions with std.dev. should be visually interpreted. Typically a shaded region communicates uncertainty: "I think the mean is somewhere within this reason, but due to statistical noise I cannot know where". Std.dev., however, is a measure of statistical variation and not uncertainty. A 95% CI would be reasonable here. However, I think this is actually an ideal place to use a tolerance interval instead, which would show both the variation and uncertainty and make the claim even more clear.

* Figure 5 would benefit from more samples in order to draw more clear differences between algorithms for Hopper, InvertedPendulum, and Walker.

* Throughout the paper uses 95% confidence intervals (which is wonderful) but frequently only 10 samples. I suspect these are percentile bootstrap confidence intervals. This needs to be stated explicitly somewhere. If these are PBCI, then likely 10 samples is insufficient to compute the confidence intervals and so these are likely to be anti-conservative. It would be useful to increase the number of samples to 30 for these instances. Perhaps Figure 6 can continue to use only 10 samples, since we can additionally lean on the structure of the sensitivity curve.

* The bolded claim in Figure 8 is probably too strong for the given evidence. It would be good to qualify the claim a little. In Acrobot, the NN may have better smoothness than either FF or FLF, it isn't clear. The only overly clear result is MountainCar followed by Cartpole.

**Strengths And Weaknesses:**

The empirical analysis is very well done. There are a few minor improvements that would help solidify claims in the paper, but overall I find the claims to be meaningful and well-supported.

Strengths:
* Fig 2b and Fig 3 make a very convincing case that FQI cannot reasonably approximate the optimal value function, no matter the size of the network. Using the proposed FLF preprocessing step and the smallest neural network, the proposed architecture learns a far more accurate value function estimate.

* The proposed FLF representation substantially reduces the computational cost of FF by eliminating all non-orthogonal feature directions---effectively forming an m-order basis. The paper then leans on neural networks to learn the (nonlinear) recombination of these bases into more complex feature spaces. This seems like a very reasonable simplification of FF and the experiments well-justify the approach.

* Fig 7 was surprising! I would have expected FF and FLF have much lower stable rank due to the fact that they have far more inputs than the standard neural network. More inputs would suggest some of the work of combining features is done in the preprocessing step and the NN is now being used more purely as a filtration (i.e. small stable rank). Instead, this result suggests that the varied features from preprocessing allows the NN to find more complex combinations of these features and retain a more rich representation (high stable rank). This is a great result.

* Table 3 is great. It is nice to see all claims tested statistically and that nearly every claim is significant.

---

**Note:** I was a reviewer for this paper at ICLR 2022 and highly appreciated the direction of the work, but felt there were several limitations of the empirical study. All of those limitations have been very well addressed.

---

> ### Author Response · Authors · 2023-05-15
>
> Dear Reviewer 1EW8,
>
> We are profoundly grateful for the considerable time and expertise you have extended in reviewing our work, not just once, but twice! To aid your evaluation, we have highlighted the significant modifications in the paper using red text.
>
> In response to your suggestions, we have revised Figure 3 to include a 95% confidence interval (CI), and we have also modified the statement associated with Figure 8.
>
> While we are fully in agreement with your suggestions to incorporate more samples in Figure 5, which could illustrate clearer differences between the Hopper, InvertedPendulum, and Walker algorithms, we regret to inform you that our current computational resources pose significant limitations. The computational demands for running these experiments are considerably high, extending over several months. We concur with your perspective that a sample size of 10 for CI computation might not provide the most accurate representation, however, we face similar resource constraints when dealing with Mujoco.
>
> As a compromise, we have prioritized increasing the training for the more conventional Gym environments and have included an additional experiment on Ant with PPO in the appendix for 30 trainings. For Ant we used Nvidia Isaac Gym Environments that are faster than the Mujoco Benchmark.
>
> We are open to further dialogue and we welcome any additional questions or comments you might have.
>
> Kind regards,
>
> The authors

---

### Decision · Action_Editors · 2023-07-07

**Recommendation:** Accept as is

**Comment:**

The authors have done a great job in addressing the concerns raised by the reviewers. The paper is well-written and easy to understand. At the end of the rebuttal there is no remaining concerns about this paper. As a result, the reviewers unanimously agreed to accept this paper.

**Audience:**

The audience of this paper would be the Deep RL community.

**Claims And Evidence:**

The claims in this paper are very well supported by the rigorous experimentations on toy and control tasks.